# VilLain: Self-Supervised Learning on Hypergraphs without Features via Virtual Label Propagation

## ABSTRACT

Group interactions arise in various scenarios in real-world systems: collaborations of researchers, co-purchases of products, and discussions in online Q&A sites, to name a few. Such higher-order relations are naturally modeled as hypergraphs, which consist of hyperedges (i.e., any-sized subsets of nodes). For hypergraphs, the challenge to learn node representation when features or labels are not available is imminent, given that (a) most real-world hypergraphs are not equipped with external features while (b) most existing approaches for hypergraph learning resort to additional information. Thus, in this work, we propose VilLain, a novel self-supervised hypergraph representation learning method based on the propagation of virtual labels (v-labels). Specifically, we learn for each node a sparse probability distribution over v-labels as its feature vector, and we propagate the vectors to construct the final node embeddings. Inspired by higher-order label homogeneity, which we discover in real-world hypergraphs, we design novel self-supervised loss functions for the v-labels to reproduce the higher-order structure-label pattern. We demonstrate that VilLain is: **(a) Requirement-free**: learning node embeddings without relying on node labels and features, **(b) Versatile**: giving embeddings that are not specialized to specific tasks but generalizable to diverse downstream tasks, and **(c) Accurate**: more accurate than its competitors for node classification, hyperedge prediction, node clustering, and node retrieval tasks.

## ACM Reference Format:
Anonymous Author(s). 2018. VilLain: Self-Supervised Learning on Hypergraphs without Features via Virtual Label Propagation. In *Proceedings of Make sure to enter the correct conference title from your rights confirmation emai (Conference acronym 'XX).* ACM, New York, NY, USA, 18 pages. https://doi.org/XXXXXXX.XXXXXXX

## 1 INTRODUCTION

In many real-world complex systems, interactions often occur in groups: research collaborations, email communications, group discussions, and protein interactions, to name a few. Representing such group interactions (i.e., higher-order relationships) as edges in an ordinary pairwise graph impairs the semantics of the interactions, often leading to considerable information loss [13, 37, 76].

Hypergraphs address the limitations of ordinary graphs by modeling group interactions as hyperedges, the non-empty subsets of

nodes. Specifically, the flexibility in hyperedge sizes enables each hyperedge to naturally represent an interaction among any number of nodes. Hypergraphs have been used to model data from various fields, including bioinformatics [31], social network analysis [73], circuit design [32], and computer vision [29, 33, 66]. Notably, hypergraph modeling has demonstrated its effectiveness over ordinary graphs in diverse applications, such as recommendation [67, 68], medical prediction [6], and crime prediction [40].

A popular approach for analyzing such complex relations is to learn node embeddings (i.e., vector representations of nodes) through *self-supervision*. In the context of hypergraphs, self-supervised learning has been applied for node classification [27, 35, 64], hyperedge prediction [62, 82], recommendation [69, 79], and user location prediction in social media [73]. Self-supervised learning enjoys several key advantages. It does not require external node labels, which are scarce in many real-world scenarios due to substantial costs in their acquisition [26]. Moreover, the learned embeddings often demonstrate considerable *versatility*, maintaining their utility across a broad range of tasks.

Many self-supervised node embedding methods require external features. Hypergraph Neural Networks (HNNs) [11, 17, 20, 28, 35, 64] and Graph Neural Networks (GNNs) [24, 34, 50, 60, 61, 70, 85], for instance, heavily rely on the external node features. As such, most of them are only tested on attributed benchmark datasets [18, 25, 49, 54, 56, 74], and their performances strongly depend on the feature quality [15, 19, 41, 46].

Despite their usefulness, external features are often entirely or partially missing in real-world hypergraphs [10, 15, 19, 53, 75, 82]. In fact, only 3.03% of the graphs at a popular graph database are given with node features [54], [1] and none of the hypergraphs at the largest hypergraph database is attributed. [2] Such a problem, in combination with the issue of label scarcity, poses an imminent challenge for hypergraph representation learning.

While some self-supervised approaches do not require external features, their embeddings are hardly versatile. Some link prediction HNNs and GNNs leverage the structural or identity features [7, 62, 78, 80, 86] without the external ones, and random walk (RW) [23, 27, 51]- or matrix factorization (MF) [47, 52, 58]-based methods (i.e. Hyper2Vec) only need graph structure for their node embeddings. However, they arguably only preserve structural properties, since their input and objective functions are *solely structural*. Such models are, thus, less applicable to tasks where the importance of structural property is less prominent, such as node classification.

Thus, in this paper, we aim to learn versatile node embeddings for hypergraphs without relying on external labels or features. To this end, we propose VilLain (**Vi**rtual **La**bel Propaga**t**ion). VilLain constructs for each node a sparse probability distribution over virtual labels (v-labels) as its feature. The probabilistic v-label assignment vectors are propagated to construct the final node embeddings.

---
[1] Out of 6,659 graph datasets, 202 are given with node attributes.
[2] https://www.cs.cornell.edu/~arb/data/

At each propagation step, the v-labels are optimized with a novel self-supervised loss function, inspired by higher-order label homogeneity in real-world hypergraphs (see Section 4). Thus, VilLain learns potential (higher-order) structure-label relationships, beyond purely structural properties.

Through extensive experiments using eight real-world hypergraphs and three downstream tasks (specifically, node classification, node retrieval, node clustering, and hyperedge prediction), we demonstrate the superiority of VilLain over 15 baseline approaches. We summarize its strengths as follows:

- **Minimum Requirements:** VilLain learns node embeddings without any supervision (e.g., node labels) or extra information (e.g., node features and the number of labels).
- **Versatile Embedding:** VilLain learns general-purpose node embeddings that are not specialized to specific tasks but generalized to diverse downstream tasks.
- **Accurate Embedding:** VilLain achieves up to 71.6%, 72.3%, and 6.7% better accuracy than unsupervised and (semi-)supervised baseline approaches for node classification, node retrieval, and hyperedge prediction tasks, respectively.

**Reproducibility.** Our code and dataset are available at https://anonymous.4open.science/r/VilLain-C18B (anonymous).

## 2 RELATED WORK

In this section, we briefly review related works on node representation learning, focusing on learning without labels or features.

**Node embedding with propagation.** Propagation has been widely applied and shown effective for both hypergraph and graph representation learning. GNNs typically have each node propagate its features to the direct neighbors [9, 22, 34], whereas for HNNs, the propagation is conducted on hypergraph structure. Specifically, HGNN [20] has each node propagate to its hyperedges, where the node feature are aggregated and propagated back to the nodes that belong to the hyperedges. HNHN [17] uses non-linear aggregation functions to update both node and hyperedge embeddings, alternately. AllSet [11] uses permutation-invariant functions to propagate on hyperedges. Other simplified GNNs [12, 16, 21, 65] first learn soft label vectors from feature vectors, which are propagated to learn the final node embeddings. Note that all the described methods require external labels or features.

**Node embedding without external labels.** Self-supervision has been widely adopted for representation learning without external labels. Self-supervised HNNs and GNNs often utilize contrastive losses. Given both original and perturbed features or structures, the models maximize the mutual information between them [35, 61, 85]. For hypergraphs, HyperGCL [64] uses node- and hyperedge-level perturbation, and TriCL [35] conducts tri-directional contrasts that maximize the agreement between two augmented views of nodes, groups, and memberships. Intuitively, such self-supervised loss functions are designed to learn node embeddings that denoise the input features and structure. It, then, implies that these self-supervised models can only learn structural properties if their input node features are random or structural.

Given random walk sequences, RW-based embedding methods [23, 27, 51] typically use Skip-Gram [44] to optimize the embeddings to maximize the likelihood of the visited nodes. MF-based approaches [47, 52, 58], on the other hand, factorize proximity matrices into low-rank matrices. As such, most RW- and MF-based embedding methods specifically preserve structural proximity.

**Node embedding without external features.** If external features are not available, HNNs and GNNs require derived features for their prediction. For structural prediction, some models have leveraged only structural information as the input features [7, 15, 62, 78, 80, 86]. Specifically, structural [4, 7, 15, 78], positional [15, 39, 63], and identity [1, 55, 62, 77, 80, 81, 86] encoding methods have been developed. Such encoding methods generally aim to enhance model expressivity beyond 1-WL test [70]. On the other hand, the majority of RW- and MF-based approaches do not require any features or labels [23, 27, 47, 51, 52, 58]. It is, however, worth noting that all the described methods *over-emphasize structural properties*, since their features and objective loss functions are solely structural. Thus, predictions from their embeddings hardly generalize to less structure-dependent tasks, such as node classification.[3]

**Relating VilLain to the prior works.** In comparison to (hyper) graph learning models without external features or labels, we present the novelty of VilLain in the subsequent sections as follows:

- **Novel Self-Supervised Loss:** Only VilLain has loss function that learns *beyond structural information* for embedding versatility.
- **Novel Input Feature Learning:** VilLain's motivation and mechanism of input feature learning are distinguished from the prior methods.

## 3 PROBLEM STATEMENT

In this section, we formulate hypergraph representation learning without features or labels. A hypergraph $G = (V, E)$ consists of a set of nodes $V = \{v_1, \cdots, v_{|V|}\}$ and a set of hyperedges $E = \{e_1, \cdots, e_{|E|}\}$. Each hyperedge $e_j \in E$ is a non-empty subset of nodes, i.e., $\varnothing \subsetneq e_j \subseteq V$. In the incidence matrix $\mathbf{H} \in \{0, 1\}^{|V| \times |E|}$ of $G$, $\mathbf{H}_{ij} = 1$, if $v_i \in e_j$, and $\mathbf{H}_{ij} = 0$ otherwise.

Given a hypergraph $G = (V, E)$, the objective of self-supervised hypergraph representation learning is to learn a node embedding $\mathbf{Z}_i \in \mathbb{R}^d$ of each node $v_i \in V$, or equivalently, a node embedding matrix $\mathbf{Z} \in \mathbb{R}^{|V| \times d}$ that captures meaningful proximity between nodes in $G$. Specifically, we aim to learn node embeddings that are generally useful for various tasks (e.g., node classification and hyperedge prediction), without relying on any kind of supervision (e.g., ground-truth semantic labels or even the number of unique labels) or external information (e.g., node attributes).

## 4 MOTIVATING OBSERVATIONS

In this section, we present our observation in real-world hypergraphs, which motivate the design of VilLain in Section 5. Inspired by pervasive *homophily* [2, 43] in real-world graphs, we postulate that hypergraphs also exhibit a similar tendency. For example, researchers from the same area tend to co-author a paper, and e-mails are likely to be exchanged within the same department. To substantiate this hypothesis, we examine label homogeneity in eight different real-world hypergraphs.

---

[3]See the low performances of such methods (e.g. Hyper2Vec, HyperGCL) in Table 2.

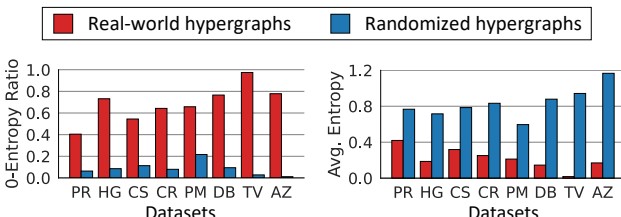

**Figure 1: Hyperedges in real-world hypergraphs (statistics in Appendix B) exhibit label homogeneity (Obs. 1).**

Using the ground-truth node labels, for each hyperedge, we measure the entropy of its soft label assignment vector, which is obtained by averaging the label assignment one-hot vectors of the nodes in the hyperedge. If the entropy is 0, all nodes in the hyperedge are labeled identically (high homogeneity). The higher the entropy is, the more diverse labels the nodes in the hyperedge have (low homogeneity). As shown in Figure 1, the entropy in real-world hypergraphs tends to be lower than that in hypergraphs that are randomized as described in [36]. Moreover, the ratio of the hyperedges with entropy 0 is much higher in real-world hypergraphs than in the randomized hypergraphs, and the average entropy is lower in real-world hypergraphs than in the randomized hypergraphs.

**OBSERVATION 1.** *Hyperedges in real-world hypergraphs exhibit label homogeneity, i.e., they tend to contain the same labeled nodes.*

In addition, we examine higher-order homogeneity in real-world hypergraphs. To this end, we measure the entropy of the higher-order label assignment vectors (or $\ell$-step labels in short) of hyperedges. For each $\ell \geq 0$, the $\ell$-step label of a hyperedge is obtained by averaging the $\ell$-step labels of the nodes in it. The $\ell$-step label of each node is given if $\ell = 0$, or obtained by averaging $(\ell - 1)$-step labels of the incident hyperedges (the detailed procedure can be found in Section 5.1). Figure 2 demonstrates that (a) the entropy of 50-step labels of hyperedges in a real-world hypergraph (spec., Trivago) is lower than those in the randomized counterpart, and (b) regardless of the step count $\ell$, hyperedges in the real-world hypergraph exhibit higher homogeneity than those in the randomized hypergraph. These findings provide concrete evidence supporting the presence of higher-order homogeneity in real-world hypergraphs. Refer to Appendix C for results from other real-world hypergraphs.

**OBSERVATION 2.** *Real-world hypergraphs exhibit higher-order label homogeneity, i.e., the node labels in each hyperedge tend to be homogeneous even after multiple steps of propagation.*

## 5 PROPOSED METHOD

In this section, we propose VilLain (Figure 3), a self-supervised node representation learning method for hypergraphs. Notably, VilLain does not require external labels or features.

### 5.1 VilLain: Virtual Label Propagation

We first present how VilLain obtains node embeddings through virtual label (v-labels) propagation, *without external features*.

**Virtual Labels.** Since node labels or features are not given, VilLain assumes the presence of $d$ v-labels and leverages the soft v-label assignment vector of each node as its learnable feature. Specifically,

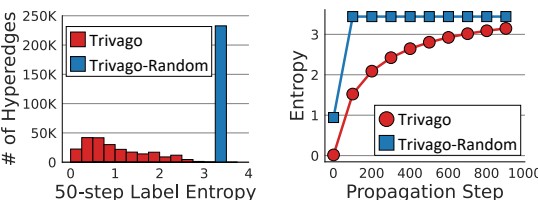

**Figure 2: Real-world hypergraphs exhibit higher-order label homogeneity (Obs. 2).**

VilLain employs a learnable matrix $\widetilde{\mathbf{X}} \in \mathbb{R}^{|V| \times d}$ where each $i^{\text{th}}$ row $\widetilde{\mathbf{X}}_i$ is used to obtain the soft assignment vector $\mathbf{X}_i^{(0)} \in [0,1]^d$ of the node $v_i$ to $d$ v-labels as follows:

$$\mathbf{X}_{ij}^{(0)} = \frac{e^{(\widetilde{\mathbf{X}}_{ij} + g_j)}}{\sum_{j'=1}^{d} e^{(\widetilde{\mathbf{X}}_{ij'} + g_{j'})}}, \quad \text{for } j = 1, \cdots, d, \quad (1)$$

where $g_j = -\log(\log(\frac{1}{u_i}))$ is random noise and $u_i \sim \text{Uniform}(0, 1)$. The above equation transforms the vector into a probability vector and encourages it to be biased towards a single v-label. As described later, the v-label assignment vectors are optimized to reproduce higher-order label homogeneity (Observations 1 and 2).

**Hypergraph V-label Propagation.** After obtaining the v-label matrix $\mathbf{X}^{(0)}$, VilLain conducts v-label propagation on the input hypergraph to obtain $\mathbf{X}^{(\ell)}$. At each step, v-labels are propagated alternately between nodes and hyperedges. Specifically, the v-label assignment matrices of hyperedges and nodes at step $\ell$ are:

$$\mathbf{Y}^{(\ell)} = \mathbf{D}_E^{-1} \mathbf{H}^T \mathbf{X}^{(\ell-1)} \quad \text{and} \quad \mathbf{X}^{(\ell)} = \mathbf{D}_V^{-1} \mathbf{H} \mathbf{Y}^{(\ell)}, \quad (2)$$

where $\mathbf{D}_V$ and $\mathbf{D}_E$ are the diagonal matrices with node degrees and hyperedge sizes, respectively. To capture higher-order dependencies among nodes, VilLain computes node embeddings $\mathbf{Z} \in [0,1]^{|V| \times d}$ by averaging the v-label assignment vectors obtained at propagation steps $1, \cdots, k'$:

$$\mathbf{Z} = \frac{1}{k'} \sum_{\ell=1}^{k'} \mathbf{X}^{(\ell)}. \quad (3)$$

Namely, the embedding $\mathbf{Z}_i$ of node $v_i$ is a probability vector averaging its v-label assignment vector at each step.

**Multi-V-label Propagation.** In real-world hypergraphs, nodes may have multiple labels, each representing different aspects. For instance, in a social network, socioeconomic status and political inclination can both serve as labels, albeit their independent homogeneity w.r.t. hypergraph topology. The same goes for the number of labels. Learning a single set of v-labels, then, can be *insufficient* to capture their complex structure-label patterns.

Thus, VilLain learns multi-v-labels for the final node embedding $\mathbf{Z}^*$. Specifically, we partition the $d$-dimensional embedding space into $D$ subspaces of potentially different dimensions, allowing for independent v-label propagation within each subspace. Then, VilLain concatenates the outputs from each subspace as follows:

$$\mathbf{Z}_i^* = \left[ \mathbf{Z}_i^{\langle 1 \rangle} \parallel \mathbf{Z}_i^{2 \rangle} \parallel \cdots \parallel \mathbf{Z}_i^{\langle D \rangle} \right], \quad (4)$$

where $\parallel$ is the concatenation operation, and $\mathbf{Z}_i^{\langle \cdot \rangle}$ is the embedding obtained from each subspace.

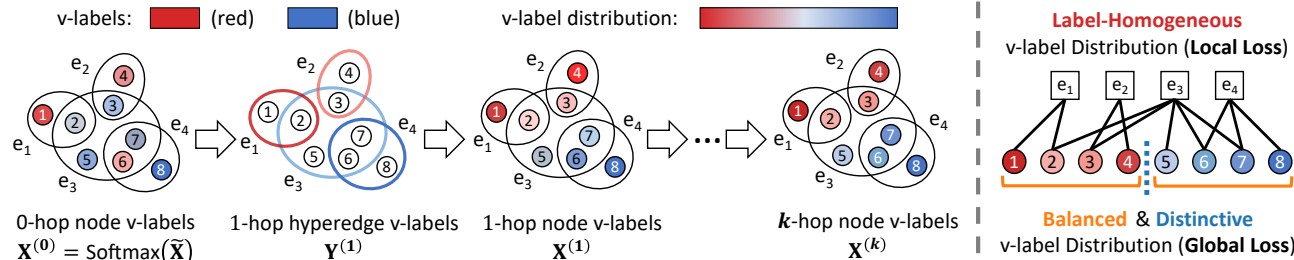

**Figure 3: (Left) Two v-labels (red and blue) are propagated between nodes and hyperedges on a hypergraph (Sec. 5.1). Note that hyperedges are colored to indicate $Y^{(1)}$. (Right) By minimizing the proposed local and global losses, the v-label distributions are learned to exhibit higher-order label homogeneity while being balanced and distinctive at each propagation step (Sec. 5.2).**

## 5.2 Self-Supervision Objectives

The learning objectives of VilLain are designed to reproduce higher-order label homogeneity by effectively capturing structural properties and also potential higher-order structure-label relationships. Recall that the entries of the matrix $\widetilde{X}$ are the only learnable parameters in VilLain that the objective function updates.

**Capturing Local Information.** Motivated by Observations 1 and 2 in Section 4, we design an objective to capture the higher-order homogeneity of nodes and hyperedges. Specifically, VilLain minimizes the entropy of the v-label assignment vectors of each node and hyperedge obtained at propagation steps $1, \cdots, k$:

$$\mathcal{L}_{\text{local}} = \sum_{\ell=1}^{k} \left( \frac{1}{|V|} \sum_{i=1}^{|V|} \mathcal{E}\left(X_i^{(\ell)}\right) + \frac{1}{|E|} \sum_{i=1}^{|E|} \mathcal{E}\left(Y_i^{(\ell)}\right) \right), \quad (4)$$

where $\mathcal{E}(p) = -\sum_i p_i \log p_i$ is the entropy measure of $p$. That is, we induce structurally close nodes (or hyperedges) to be assigned to the same v-label. Beyond capturing the homogeneity at the hyperedge level, i.e., $\ell = 1$ (Observation 1), the loss function is designed to reproduce the higher-order homogeneity of nodes and hyperedges by minimizing the entropy of v-label assignment vectors at each propagation step $\ell \in \{1, \cdots, k\}$ (Observation 2). For training speed, the number of steps $k$ for training can be smaller than $k'$ for inference.

**Capturing Global Information.** VilLain also considers the global distribution of labels. To this end, we give v-label-level supervision to VilLain so that v-labels are properly distributed over the entire hypergraph. First, since Eq. (4) is trivially minimized when all nodes and hyperedges are assigned to a single v-label, we use the following term to prevent this problem:

$$\mathcal{J}_{\text{cls}} = -\sum_{\ell=1}^{k} \left( \mathcal{E}\left(x^{(\ell)}\right) + \mathcal{E}\left(y^{(\ell)}\right) \right) \quad (5)$$

$$\text{where} \quad x_i^{(\ell)} = \frac{\|X_{:,i}^{(\ell)}\|_1}{\sum_{j=1}^{d} \|X_{:,j}^{(\ell)}\|_1} \quad \text{and} \quad y_i^{(\ell)} = \frac{\|Y_{:,i}^{(\ell)}\|_1}{\sum_{j=1}^{d} \|Y_{:,j}^{(\ell)}\|_1}.$$

Here, $x^{(\ell)} = [x_1^{(\ell)}, \cdots, x_d^{(\ell)}]$ and $y^{(\ell)} = [y_1^{(\ell)}, \cdots, y_d^{(\ell)}]$ denote the weighted ratios of nodes and hyperedges for each v-label at step $\ell$. Note that $X_{:,i}^{(\ell)}$ and $Y_{:,i}^{(\ell)}$, which are the $i$th columns of $X^{(\ell)}$ and $Y^{(\ell)}$, correspond to the vectors of v-label $i$ for nodes and hyperedges,

respectively. That is, we maximize the *entropy of the global distribution* of the v-labels at each step, restraining any single v-label from dominating the entire hypergraph.

In addition, we aim to make v-labels distinctive by making the sets of nodes and hyperedges assigned to each v-label nearly disjoint from those with another v-label. To this end, we minimize the following cross-entropy-based objective:

$$\mathcal{J}_{\text{dst}} = -\sum_{\ell=1}^{k} \sum_{i=1}^{d} \left( \log \bar{x}_i^{(\ell)} + \log \bar{y}_i^{(\ell)} \right) \quad (6)$$

$$\text{where} \quad \bar{x}_i^{(\ell)} = \frac{e^{\mathcal{S}\left(X_{:,i}^{(\ell)}, X_{:,i}^{(\ell)}\right)}}{\sum_{j=1}^{d} e^{\mathcal{S}\left(X_{:,i}^{(\ell)}, X_{:,j}^{(\ell)}\right)}} \quad \text{and} \quad \bar{y}_i^{(\ell)} = \frac{e^{\mathcal{S}\left(Y_{:,i}^{(\ell)}, Y_{:,i}^{(\ell)}\right)}}{\sum_{j=1}^{d} e^{\mathcal{S}\left(Y_{:,i}^{(\ell)}, Y_{:,j}^{(\ell)}\right)}}.$$

Here, $\bar{x}_i^{(\ell)}$ and $\bar{x}_j^{(\ell)}$ indicate the distinctiveness of v-label $i$ at each propgation step $\ell$ in nodes and hyperedges, respectively, and $\mathcal{S}(\cdot, \cdot)$ measures the cosine similarity of two input vectors. Minimizing Eq. (6) reinforces the *distinctiveness of each v-label* at each step.

Finally, we minimize the global-level loss, defined as the sum of Eq. (5) and Eq. (6), to let v-labels be properly distributed across the entire hypergraph:

$$\mathcal{L}_{\text{global}} = \mathcal{J}_{\text{cls}} + \mathcal{J}_{\text{dst}} \quad (7)$$

**Objective Function.** To exhibit both the local and global structure-label patterns, VilLain minimizes both objectives, Eq. (4) and (7):

$$\mathcal{L} = \mathcal{L}_{\text{local}} + \mathcal{L}_{\text{global}}.$$

While we can introduce a hyperparameter for balancing $\mathcal{L}_{\text{local}}$ and $\mathcal{L}_{\text{global}}$, we simply add the two losses since hyperparameter tuning based on external supervision is strictly restricted in our setting.

Note that, by reproducing higher-order label homogeneity, VilLain captures not only structural properties but also potential higher-order structure-label relationships. Consequently, compared to self-supervised methods that exclusively focus on structural aspects (see Section 2 for further discussion), VilLain can learn effective embeddings for less structure-dependent tasks, such as node classification, as confirmed empirically in Section 6.

**Complexity Analysis.** We analyze the time and space complexity of VilLain for computing the final embedding $Z^*$, as well as the computational cost associated with optimizing their losses. Specifically,

when the dimension of each subspace is $d/D$, it takes:

$$O\left(kd \sum_{e \in E} |e| + \frac{kd^2}{D}(|V| + |E|)\right) \text{ time and } O\left(\frac{kd^2}{D}(|V| + |E|)\right) \text{ space}$$

for propagating v-labels (Eq. (2)) and computing losses $\mathcal{L}_{\text{local}}$ (Eq. (4)), $\mathcal{J}_{\text{cls}}$ (Eq. (5)), and $\mathcal{J}_{\text{dst}}$ (Eq. (6)) for $1, \cdots, k$ steps. To generate node embeddings (Eq. (3)), the losses are not necessarily computed, and thus it takes $O(k'd \sum_{e \in E} |e|)$ time and requires $O(k'd(|V| + |E|))$ space. The details can be found in Appendix A. Importantly, introducing multi-v-labels (i.e., $D > 1$) leads to the reduction in time and space complexity, thereby indicating an additional advantage of learning v-labels in multiple subspaces. This is empirically supported in Section 6.4.

## 5.3 Extension to Unobserved Nodes

Heretofore, we described how VilLain learns node embeddings $\mathbf{Z}$ from a static hypergraph. However, in many scenarios, hypergraphs evolve over time (e.g., new members in the group), introducing new nodes and hyperedges to the hypergraph. This motivates us to extend VilLain to generate embeddings also for newly introduced, unobserved nodes and hyperedges. In this subsection, we extend VilLain to embed such unobserved nodes and hyperedges.

**Settings.** Consider a connected hypergraph $G_S = (V_S, E_S)$, which is a subset of a connected hypergraph $G = (V, E)$, where $V_S \subseteq V$ and $E_S \subseteq E$. Using the incidence matrix $\mathbf{H}_S \in \{0, 1\}^{|V_S| \times |E_S|}$ of $G_S$, VilLain has generated v-labels and embeddings $\mathbf{X}_S^{(0)}, \mathbf{Z}_S \in \mathbb{R}^{|V_S| \times d}$, respectively, for the observed nodes $V_S$. Nodes $V \setminus V_S$ and hyperedges $E \setminus E_S$ are introduced after VilLain training.

**Embedding Unobserved Nodes.** To embed nodes including the unobserved ones $V \setminus V_S$, VilLain propagates learned v-labels $\mathbf{X}_S^{(0)}$ of the observed nodes $V_S$ on hypergraph $G$ containing the unobserved nodes and hyperedges. Specifically, v-label assignment matrices for all nodes $\mathbf{X}^{(\ell)}$ and hyperedges $\mathbf{Y}^{(\ell)}$ at step $\ell \geq 1$ are obtained like in Eq. (2) as follows:

$$\mathbf{Y}^{(\ell)} = \mathbf{D}_E^{-1} \mathbf{H}^T \mathbf{X}^{(\ell-1)} \quad \text{and} \quad \mathbf{X}^{(\ell)} = \mathbf{D}_V^{-1} \mathbf{H} \mathbf{Y}^{(\ell)},$$

where $\mathbf{X}^{(0)} \in \mathbb{R}^{|V| \times d}$ is $\mathbf{X}_S^{(0)}$ with zero-paddings at row indices of the nodes $V \setminus V_S$. Since we assume a connected hypergraph $G$, there always exists $\ell'$ such that all nodes $V$ are assigned non-zero v-labels. Then, using Eq. (3), $\mathbf{X}^{(\ell')}, \cdots, \mathbf{X}^{(k')}$ are used to generate embeddings $\mathbf{Z}$ for all nodes $V$, where $k' \geq \ell'$. We empirically show that VilLain generates informative embeddings for unobserved nodes in Section 6.4.

## 6 EXPERIMENTAL RESULTS

In this section, we present experimental results for four downstream tasks utilizing node embeddings. We first assess the accuracy of VilLain by comparing it with the state-of-the-art (hyper)graph representation learning methods (Section 6.2). Then, we demonstrate the effectiveness of each design choice of VilLain (Section 6.3). Lastly, we conduct additional analyses on VilLain (Section 6.4).

## 6.1 Experimental Settings

In this subsection, we report the experimental settings.

**Table 1: Summary statistics of eight real-world hypergraphs: the number of nodes $|V|$, the number of hyperedges $|E|$, the size of the hypergraph $\sum_{e \in E} |e|$, the number of edges $|\mathcal{E}|$ in the clique expansion, and the number of ground-truth labels.**

| Dataset | | $|V|$ | $|E|$ | $\sum_{e \in E} |e|$ | $|\mathcal{E}|$ | # Labels |
|---|---|---|---|---|---|---|
| Primary (PR) | [57] | 242 | 12,704 | 30,729 | 8,317 | 11 |
| High (HG) | [42] | 327 | 7,818 | 18,192 | 5,818 | 9 |
| Citeseer (CS) | [71] | 1,019 | 819 | 2,808 | 3,867 | 6 |
| Cora (CR) | [71] | 1,330 | 1,503 | 4,599 | 4,144 | 7 |
| Pubmed (PM) | [71] | 3,824 | 7,951 | 34,605 | 123,819 | 3 |
| DBLP (DB) | [71] | 36,188 | 18,924 | 90,868 | 425,669 | 6 |
| Trivago (TV) | [14] | 172,738 | 233,202 | 726,861 | 1,095,204 | 160 |
| Amazon (AZ) | [45] | 260,209 | 31,964 | 422,076 | 14,142,811 | 10 |

**Datasets.** We use eight publicly available real-world hypergraphs summarized in Table 1. All datasets are derived from group interactions that arise in real-world scenarios (e.g., coauthorship and co-purchase). For details regarding the preprocessing method and descriptions for each dataset, refer to Appendix B.1.

**Baselines.** We consider 15 unsupervised and (semi-)supervised graph and hypergraph embedding methods as competitors. Deepwalk [51], Node2vec [23], DGI [61], GRACE [85], GMI [50], Hyper2vec [27], LBSN [73], and TriCL [35] are unsupervised methods, and GCN [34], GAT [60], HGNN [20], HNHN [17], AllSet [11], UniGNN [28], and HyperGCL [64] are (semi-)supervised methods. For graph embedding methods (i.e., GCN, GAT, Deepwalk, Node2vec, DGI, GRACE, and GMI), we use the clique expansion of the hypergraph.[4] For all methods that require node features (i.e., GCN, GAT, DGI, GRACE, GMI, HGNN, HNHN, AllSet, UniGNN, HyperGCL, and TriCL), we use the embeddings obtained by Hyper2vec,[5] which lead to the best performance among three alternatives (see Appendix C for detailed results).

**Implementations.** We simply use $k = 4$ for VilLain and all its variants and use $k' = 10$ for small datasets (s.t., $|V| < 10,000$) and $k' = 100$ for large datasets (s.t., $|V| \geq 10,000$). As discussed in Section 5.1, to capture diverse structural-label information, we aggregate embeddings obtained with various numbers of v-labels. Specifically, we concatenate embeddings obtained using different numbers of v-labels. For each number $\lceil \frac{d}{D} \rceil \in \{2, 3, \cdots, 8\}$ of v-labels, we learn $D$ subspaces and then perform PCA to ensure that the final embedding is of the target dimension $d$. Refer to Appendix B.2 for the detailed settings of other baselines.

## 6.2 Accuracy of VilLain

To verify the quality of the VilLain's node embeddings, we consider four downstream tasks on hypergraphs: node classification, node retrieval, node clustering, and hyperedge prediction. The embedding dimension of all methods, including VilLain, is fixed to 128. Results including standard deviation is provided in Appendix C.2.

**Node Classification.** We perform node classification by randomly and disjointly splitting the dataset into training, validation, and test sets. For training and validation sets, the labels of 20 nodes per class are given for all datasets except for Primary and High, where the labels of 2 nodes are given per class. The remaining

---

[4]The clique expansion is the pairwise graph obtained by replacing each hyperedge with the clique formed by the nodes in the hyperedge.

[5]For Amazon, we use Node2vec since Hyper2vec ran out of time (> 24 hours).

**Table 2: VilLain performs best on node classification in terms of accuracy. Each baseline method is designed for either graphs or hypergraphs and for either semi-supervised or unsupervised settings.**

| Method | DBLP | Trivago | Amazon | Primary | High | Citeseer | Cora | Pubmed | Rank |
|---|---|---|---|---|---|---|---|---|---|
| GCN | 67.37 ± 1.45 | 38.06 ± 1.49 | 28.73 ± 4.73 | 75.63 ± 5.08 | 96.25 ± 2.55 | 60.64 ± 3.47 | 72.96 ± 1.82 | 77.56 ± 2.58 | 7.00 ± 2.91 |
| GAT | 61.74 ± 1.97 | 51.52 ± 0.68 | 30.94 ± 2.13 | 66.79 ± 4.73 | 90.58 ± 2.76 | 49.57 ± 2.64 | 58.09 ± 2.14 | 73.67 ± 1.78 | 11.75 ± 3.83 |
| Deepwalk | 29.03 ± 1.43 | 16.85 ± 0.45 | 25.43 ± 1.72 | 84.89 ± 3.67 | 99.31 ± 0.48 | 45.10 ± 3.18 | 56.58 ± 1.88 | 68.58 ± 2.60 | 11.62 ± 4.71 |
| Node2vec | 29.21 ± 1.89 | 16.88 ± 0.44 | 25.27 ± 2.36 | 83.53 ± 3.09 | **99.38 ± 0.45** | 45.37 ± 3.17 | 59.15 ± 1.84 | 69.05 ± 3.00 | 11.00 ± 4.35 |
| DGI | 62.37 ± 3.32 | 73.46 ± 1.22 | 31.80 ± 1.45 | 86.66 ± 4.51 | 92.49 ± 0.60 | 61.36 ± 2.91 | 71.23 ± 2.04 | 77.51 ± 1.38 | 7.25 ± 3.59 |
| GRACE | 71.86 ± 2.51 | OOM | OOM | 63.78 ± 5.12 | 99.03 ± 0.30 | 61.16 ± 2.78 | 73.43 ± 1.81 | 77.70 ± 1.81 | 5.50 ± 4.75 |
| GMI | 64.19 ± 1.63 | OOM | OOM | 80.10 ± 4.94 | 96.61 ± 2.63 | 58.67 ± 2.68 | 71.31 ± 1.69 | 75.51 ± 2.77 | 9.16 ± 1.57 |
| HGNN | 66.60 ± 2.18 | OOM | OOM | 88.28 ± 5.02 | 92.19 ± 3.84 | 60.91 ± 2.32 | 72.90 ± 2.00 | 76.58 ± 2.86 | 7.50 ± 3.09 |
| HNHN | 63.99 ± 2.21 | 59.52 ± 1.64 | 28.99 ± 2.63 | 91.31 ± 2.47 | 96.83 ± 1.25 | 59.02 ± 1.63 | 68.81 ± 1.26 | 75.33 ± 1.77 | 7.50 ± 2.39 |
| AllSet | 63.67 ± 1.89 | 36.58 ± 0.93 | 21.75 ± 1.67 | 85.94 ± 3.02 | 95.70 ± 1.66 | 56.08 ± 1.95 | 67.73 ± 1.81 | 74.11 ± 2.04 | 10.75 ± 1.08 |
| UniGNN | 67.16 ± 2.15 | 69.98 ± 1.60 | 33.77 ± 3.22 | 88.88 ± 3.58 | 95.12 ± 3.97 | 59.10 ± 2.76 | 71.44 ± 1.03 | 74.37 ± 2.10 | 7.12 ± 3.09 |
| HyperGCL | 58.72 ± 1.54 | 74.99 ± 1.23 | 22.86 ± 2.01 | 74.07 ± 6.06 | 85.79 ± 8.92 | 57.54 ± 1.61 | 74.99 ± 1.33 | 78.44 ± 3.33 | 8.87 ± 5.18 |
| Hyper2vec | 67.18 ± 1.78 | 75.82 ± 1.45 | OOT | 92.52 ± 2.45 | 96.34 ± 1.34 | 61.50 ± 2.60 | 71.79 ± 1.63 | 77.04 ± 1.51 | 4.85 ± 2.35 |
| LBSN | 22.63 ± 2.20 | 47.99 ± 0.82 | 11.56 ± 0.90 | 86.71 ± 3.71 | 95.87 ± 2.28 | 45.43 ± 2.15 | 59.70 ± 1.31 | 54.89 ± 2.38 | 11.87 ± 3.21 |
| TriCL | 68.18 ± 1.36 | OOM | OOM | 92.67 ± 2.50 | 98.10 ± 1.02 | 59.17 ± 3.35 | 72.35 ± 1.53 | 78.57 ± 1.88 | 4.16 ± 1.95 |
| **VilLain** | **77.16 ± 1.26** | **79.43 ± 1.63** | **57.95 ± 2.47** | **93.66 ± 3.93** | 99.19 ± 0.41 | **61.53 ± 3.17** | **75.03 ± 1.38** | **78.82 ± 1.47** | **1.25 ± 0.66** |

**Table 3: VilLain performs overall best on hyperedge prediction (in terms of accuracy), node clustering (in terms of normalized mutual information), and node retrieval (in terms of mean average precision).**

| Method | Hyperedge Prediction (Acc.) | | | | | | | | | Node Clustering (NMI) | | | | | | | | | Node Retrieval (MAP) | | | | | | | | |
|---|---|---|---|---|---|---|---|---|---|---|---|---|---|---|---|---|---|---|---|---|---|---|---|---|---|---|---|
| | DB | TV | AZ | PR | HG | CS | CR | PM | Rank | DB | TV | AZ | PR | HG | CS | CR | PM | Rank | DB | TV | AZ | PR | HG | CS | CR | PM | Rank |
| Deepwalk | 63.9 | 61.3 | 69.4 | 83.8 | 85.9 | 69.6 | 67.2 | 65.9 | 6.25 | 0.7 | 16.7 | 7.8 | 85.2 | **100.0** | 14.6 | 23.9 | **34.4** | 5.00 | 21.3 | 7.5 | 27.7 | 81.6 | 98.7 | 27.6 | 29.2 | 49.0 | 6.37 |
| Node2vec | 64.2 | 61.4 | 69.3 | 83.2 | 85.4 | 70.4 | 66.9 | 65.8 | 6.62 | 0.9 | 17.0 | 7.7 | 83.5 | **100.0** | 14.5 | 23.8 | 32.8 | 5.87 | 21.6 | 7.1 | 27.8 | 81.1 | 98.6 | 27.3 | 29.4 | 49.4 | 6.50 |
| DGI | **86.1** | 83.8 | 90.8 | 79.1 | 84.4 | 79.2 | 76.3 | 80.9 | 3.75 | 16.6 | 44.5 | 13.0 | 84.4 | 73.9 | 29.1 | 32.1 | 31.3 | 5.62 | 36.1 | 37.3 | 31.1 | 89.7 | 97.8 | 43.8 | 50.6 | 61.7 | 3.25 |
| GRACE | 85.4 | OOM | OOM | 80.3 | 87.4 | 77.9 | 74.5 | 79.1 | 4.00 | 43.0 | OOM | OOM | 67.6 | 98.2 | 33.0 | 46.0 | 31.6 | 5.00 | 50.2 | OOM | OOM | 61.4 | **99.5** | 41.1 | 54.2 | 60.9 | 4.00 |
| GMI | 75.6 | OOM | OOM | 82.4 | 85.9 | 74.4 | 69.4 | 72.3 | 5.50 | 27.8 | OOM | OOM | 84.1 | 93.1 | 25.3 | 42.6 | 18.7 | 6.50 | 34.6 | OOM | OOM | 80.0 | 97.8 | 35.9 | 41.6 | 55.1 | 6.33 |
| Hyper2vec | 71.2 | 72.4 | OOT | 76.4 | 79.6 | 78.1 | 71.7 | 71.5 | 6.14 | 43.4 | 66.3 | OOT | **92.5** | 99.3 | 34.3 | 45.5 | 33.6 | **2.27** | 35.5 | 43.1 | OOT | 85.7 | 90.7 | 41.2 | 46.7 | 55.6 | 4.85 |
| LBSN | 48.7 | 89.1 | 63.7 | 79.4 | 87.1 | 74.3 | 69.6 | 66.1 | 5.87 | 1.1 | 39.4 | 2.7 | 85.5 | 97.8 | 12.1 | 29.0 | 4.6 | 6.50 | 21.0 | 19.1 | 29.1 | 81.3 | 93.2 | 30.6 | 40.1 | 43.5 | 6.62 |
| TriCL | 77.4 | OOM | OOM | **84.0** | 87.8 | 82.0 | 76.7 | 80.5 | 2.33 | 38.0 | OOM | OOM | 87.8 | 98.7 | 34.4 | 44.8 | 33.7 | 3.00 | 45.1 | OOM | OOM | 89.9 | 97.6 | 42.4 | 55.0 | 61.9 | 3.16 |
| **VilLain** | 81.6 | **95.1** | **94.9** | 83.2 | **87.8** | **82.1** | **79.0** | **82.8** | **1.50** | **46.6** | **69.4** | **35.2** | 85.7 | 98.7 | **34.5** | **50.4** | 32.7 | **2.25** | **60.2** | **67.2** | **53.6** | **91.3** | 99.0 | **46.4** | **58.0** | **64.4** | **1.12** |

nodes are used as the test set. For un- or self-supervised methods including VilLain, we evaluate the accuracy of logistic regression using the embeddings obtained from each method. Table 2 shows the accuracy of all methods in all datasets. VilLain ranks first on average, showing the best performance. We conjecture that v-label propagation inherits rich structural properties and also potential higher-order structure-label relationships, generating high-quality representations of nodes.

**Hyperedge Prediction.** The problem of hyperedge prediction is formulated as a binary classification task, predicting whether the given hyperedge is real or fake [30, 48, 76]. Given a set $E$ of real hyperedges, we generate a set $E'$ of fake hyperedges with the same hyperedge size distribution by randomly sampling subsets of nodes. To obtain the embedding of each hyperedge, we apply maxmin pooling [6] to the embeddings of the nodes in it. For more training details on hyperedge prediction, refer to Appendix B.4. As shown in Table 3, VilLain performs the best on average. We conjecture that VilLain, which captures potential structure-label relations, is effective for this task because it indirectly relates to labels due to the high label homogeneity of real hyperedges.

**Node Clustering.** For the clustering task, we group nodes into the number of unique ground-truth labels, applying k-means to the

learned embeddings. Then, we compute the Normalized Mutual Information (NMI) to assess the quality of clustering. As shown in Table 3, VilLain outperforms all baseline methods in terms of average ranks. This indicates that the embeddings learned by VilLain exhibit meaningful semantic similarities in their distribution.

**Node Retrieval.** The problem of node retrieval aims to search for similar nodes of a given query node, using the learned embeddings. Specifically, we retrieve nodes based on the cosine similarity between their embeddings and the embedding of the query node. Then, we compute the Mean Average Precision (MAP), to measure the retrieval quality. Intuitively, the retrieval is considered to be successful if the nodes of the same class as the query node are highly ranked. For more details regarding the task, refer to Appendix B.3. As shown in Table 3, VilLain outperforms baseline methods, with a large margin. These results imply that v-labels, which are *virtual* and learned without any ground-truth node labels, are useful for finding similar nodes of the same class.

## 6.3 Ablation Study

In this subsection, we conduct ablation studies to verify the effectiveness of each component of VilLain by comparing its performance to that of its variants.

---

[6]We compute maxmin pooling by: elementwise max pooling - elementwise min pooling. An alternative pooling method is compared in Appendix C.8.

**Table 4: VilLain outperforms its three variants, VilLain-S, VilLain-M, and VilLain-L, in four downstream tasks, implying that VilLain benefits from (1) propagating v-labels in multiple subspaces, (2) aggregating embeddings from various numbers of v-labels, and (3) reproducing both local and global structure-label patterns for self-supervision.**

| Method | Node Classification (Accuracy) | | | | | | | | Hyperedge Prediction (Accuracy) | | | | | | | | Node Clustering (NMI) | | | | | | | | Node Retrieval (MAP) | | | | | | | |
|---|---|---|---|---|---|---|---|---|---|---|---|---|---|---|---|---|---|---|---|---|---|---|---|---|---|---|---|---|---|---|---|---|
| | DB | TV | AZ | PR | HG | CS | CR | PM | DB | TV | AZ | PR | HG | CS | CR | PM | DB | TV | AZ | PR | HG | CS | CR | PM | DB | TV | AZ | PR | HG | CS | CR | PM |
| VilLain-S | 69.5 | OOM | OOM | 83.1 | 96.8 | 59.9 | 71.0 | 77.1 | 79.7 | OOM | OOM | 79.2 | 86.4 | 80.9 | 75.4 | 78.9 | 35.7 | OOM | OOM | **88.4** | 97.9 | 21.3 | 30.3 | 25.8 | 45.3 | OOM | OOM | 65.3 | 98.4 | 41.4 | 47.2 | 60.9 |
| VilLain-M | 74.2 | 75.1 | 54.8 | _91.6_ | _98.6_ | 61.1 | 73.7 | _78.5_ | 80.7 | _95.0_ | _94.7_ | _83.0_ | _87.5_ | _82.4_ | **79.0** | _82.6_ | _43.5_ | 65.4 | _35.3_ | _87.3_ | **98.8** | 31.9 | _46.2_ | _34.7_ | 49.2 | 46.7 | 50.4 | _86.1_ | _98.7_ | 44.0 | 53.7 | 62.1 |
| VilLain-L | _76.9_ | _79.3_ | _56.7_ | 64.5 | 97.6 | **61.9** | _74.1_ | 78.1 | _81.4_ | 94.9 | 94.2 | 76.6 | 87.4 | **82.9** | 78.8 | 82.1 | 42.7 | _66.6_ | **36.2** | 64.0 | 96.6 | **37.1** | 43.9 | _33.0_ | _59.3_ | _66.1_ | _51.6_ | 66.1 | 97.5 | **46.5** | _54.9_ | _63.3_ |
| VilLain | **77.2** | **79.4** | **58.0** | **93.7** | **99.2** | _61.5_ | **75.0** | **78.8** | **81.6** | **95.1** | **94.9** | **83.2** | **87.8** | 82.1 | **79.0** | **82.8** | **46.6** | **69.4** | 35.2 | 85.7 | _98.7_ | _34.5_ | **50.4** | 32.7 | **60.2** | **67.2** | **53.6** | **91.3** | **99.0** | _46.4_ | **58.0** | **64.4** |

**Table 5: VilLain benefits from the long-range propagation of v-labels. Increasing both the number of v-label propagation ($k$ for loss computation and $k'$ for embedding generation) tends to improve the node classification accuracy.**

| | DB | TV | AZ | PR | HG | CS | CR | PM | Rank |
|---|---|---|---|---|---|---|---|---|---|
| $k=1$ | 74.25 | 78.14 | 52.16 | **94.67** | **99.51** | 60.48 | 74.96 | 78.21 | 3.00 |
| $k=2$ | 75.76 | 78.44 | 55.09 | 93.43 | _99.29_ | 60.17 | **75.15** | _78.97_ | 2.62 |
| $k=4$ | _77.16_ | _79.43_ | _57.95_ | _93.66_ | 99.19 | _61.53_ | _75.03_ | 78.82 | _2.25_ |
| $k=8$ | **78.22** | **80.24** | **59.12** | 92.47 | 98.78 | **62.05** | 74.24 | **79.22** | **2.12** |
| $k'=1$ | 64.71 | 60.60 | 48.46 | **96.74** | **99.58** | 60.62 | 74.68 | 77.94 | 5.62 |
| $k'=2$ | 65.29 | 61.59 | 49.42 | _96.36_ | _99.57_ | 60.44 | 74.70 | 78.18 | 5.50 |
| $k'=4$ | 66.64 | 63.25 | 50.52 | 96.33 | 99.39 | 60.54 | 74.77 | 78.29 | 5.00 |
| $k'=8$ | 67.88 | 65.08 | 53.22 | 93.91 | 99.26 | 61.29 | **75.06** | 78.75 | 4.50 |
| $k'=16$ | 70.83 | 68.28 | 54.65 | 94.49 | 98.86 | 61.62 | _74.86_ | 79.12 | _3.75_ |
| $k'=32$ | 73.20 | 72.31 | 55.80 | 92.57 | 98.59 | 61.96 | 74.68 | _79.22_ | 4.00 |
| $k'=64$ | _76.47_ | _76.77_ | _56.42_ | 94.21 | 98.50 | _62.42_ | 74.25 | 78.98 | 4.00 |
| $k'=128$ | **77.62** | **80.63** | **57.46** | 88.68 | 98.09 | **63.67** | 74.41 | **79.37** | **3.50** |

**Table 6: VilLain yields informative embeddings even for unobserved nodes. Fully observed hypergraphs consist of the entire set $V$ of nodes, whereas partially observed hypergraphs only contain the subset $V_S \subseteq V$ of nodes after removing 50% of the hyperedges. Despite a performance decrease compared to its fully observable settings, VilLain outperforms its strongest baseline, TriCL in node classification, even for the set $V \setminus V_S$ of nodes are not observed in VilLain but observed in TriCL during training.**

| Learning/Node Type | | | DB | TV | AZ | CS | CR | PM |
|---|---|---|---|---|---|---|---|---|
| Fully Observed | VilLain | $V_S$ | 78.01 | 80.03 | 56.77 | 62.75 | 75.43 | 79.21 |
| | | $V \setminus V_S$ | 66.19 | 76.07 | 58.74 | 57.52 | 73.46 | 69.62 |
| Partially Observed | VilLain | $V_S$ | 76.45 | 78.66 | 53.72 | 62.49 | 73.79 | 77.78 |
| | | $V \setminus V_S$ | 65.86 | 74.71 | 55.23 | 56.42 | 72.27 | 69.71 |
| Fully Observed | TriCL | $V_S$ | 69.20 | OOM | OOM | 60.83 | 72.94 | 78.99 |
| | | $V \setminus V_S$ | 55.11 | OOM | OOM | 53.74 | 70.02 | 68.60 |

**Effectiveness of Multi-V-label Learning.** To demonstrate the effectiveness of using multiple subspaces, we consider two variants of VilLain: (a) **VilLain-S** learns $d$ v-labels in a single embedding space and (b) **VilLain-M** learns $\lceil d/D \rceil$ v-labels in $D$ subspaces. In Table 4, we compare VilLain with its two variants on the four considered tasks. Regarding VilLain-M, we report the average accuracy when $\lceil d/D \rceil = \{2, 3, \cdots, 8\}$. We first observe that VilLain-M consistently outperforms VilLain-S, indicating the effectiveness of the multi-v-label propagation. Additionally, introducing multiple subspaces enhances the space complexity, as VilLain-M avoids out-of-memory issues in large hypergraphs like Amazon and Trivago, in contrast to VilLain-S. This aligns with our space complexity analysis presented in Section 5. Furthermore, the superior performance of VilLain over VilLain-M implies that aggregating embeddings from various numbers of v-labels (see Section 6.1 for details) captures more informative potential structure-label relations.

**Effectiveness of Loss Functions.** To examine the effectiveness of the designed loss functions, we consider another variant of VilLain, **VilLain-L**, which only uses the local loss $\mathcal{L}_{\text{local}}$ to learn v-label distributions. As shown in Table 4, VilLain, which jointly optimizes $\mathcal{L}_{\text{local}}$ and $\mathcal{L}_{\text{global}}$ and thus captures both local and global information of the input hypergraph, outperforms VilLain-L, demonstrating the effectiveness of the proposed loss functions. In Appendix C.3, we analyze when $\mathcal{L}_{\text{global}}$ is particularly beneficial.

**Effects of Long-Range V-label Propagation.** To examine the effects of the long-range propagation of v-labels, we test how the

number of steps $k$ (during loss computation) and $k'$ (during embedding generation) affect the performance of VilLain in node classification. As shown in Table 5, except for Primary and High, which are the smallest datasets, adopting long-range propagation of v-labels is beneficial. In particular, we can see that large datasets (e.g., DBLP, Trivago, and Amazon) benefit from large $k$s and $k'$s. This tendency holds in other tasks (i.e., hyperedge prediction, node clustering, and node retrieval) as shown in Appendix C.6. This implies that the higher-order label homogeneity, which VilLain aims to reproduce, positively affects the performance in downstream tasks.

## 6.4 Further Analysis of VilLain

In this subsection, we summarize additional experimental results, a part of which is provided in Appendices C and D. Here, we consider the node classification task for evaluation, unless otherwise stated.
**Scalability of VilLain.** We test the scalability of VilLain by measuring its training time. In order to test scalability on larger hypergraphs, we upscale Cora using HyperCL [36] by $2^{\{5.0, 5.5, \cdots, 8.0\}}$ times. As seen in Figure 4, VilLain scales linearly with the size of the hypergraph and also the number of propagation steps. In addition, the training time decreases with an increased number of subspaces, which is consistent to our time complexity analysis in Section 5.
**Performances on Unobserved Nodes.** In Section 5.3, we discussed how VilLain can generate embeddings for nodes that are not observed during training. Instead of using the original hypergraph $G = (V, E)$, we evaluate how VilLain, after learning embeddings for the subset $V_S$ of nodes from a partial hypergraph $G_S = (V_S, E_S)$, effectively generates node embeddings for both sets $V_S$ and $V \setminus V_S$

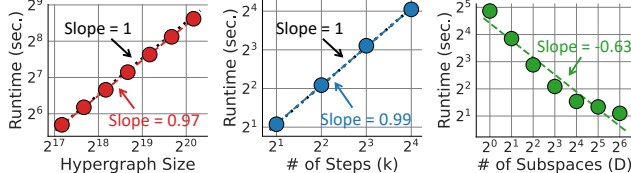

**Figure 4: The training time (for 100 epochs) of VilLain is linear in the hypergraph size (i.e., $\sum_{e \in E} |e|$) and the number of steps of v-label propagation (i.e., $k$). The training time decreases with respect to the number $D$ of subspaces, implying the efficiency of multi-space v-label propagation which is consistent with the complexity analysis in Section 5.**

of nodes. Indeed, due to the utilization of reduced structural information, it is natural to expect a degraded quality of node embeddings for both $V_S$ and $V \setminus V_S$ sets of nodes in this scenario. This degradation is empirically shown in Table 6 in comparison to the fully-observable setting. However, VilLain outperforms its strongest baseline, TriCL, across six datasets,[7] even when utilizing partial hypergraphs with 50% of hyperedges removed. TriCL, on the other hand, employs complete hypergraphs to learn embeddings for both sets of nodes. This demonstrates the effectiveness of VilLain in generating informative embeddings for unobserved nodes, as well as its robustness to the removed hyperedges.

**Performance on Less Homophilic Hypergraphs.** While VilLain is rooted in the insights gained from the observations of higher-order label homogeneity across various real-world hypergraphs (refer to Section 4), it demonstrates a comparable level of performance also in less homophilic hypergraphs. In Figure 5, we generated semi-real hypergraphs by (1) selecting two hyperedges uniformly at random, and (2) interchanging a single node from each. We repeat this process $\{100, 200, \cdots, 1000\}$ and $\{1000, 2000, \cdots, 10000\}$ times in Cora and DBLP, respectively, resulting in hypergraphs with a diverse range of increased hyperedge entropy (i.e., heterophilicity) and thus less homophilic. From the results, we can observe that the node classification accuracies of VilLain in Cora and DBLP degrade with the degree of heterophilicity in the hypergraph. Nonetheless, its performance remains superior to that of the two strongest baselines, Hyper2vec and TriCL, demonstrating its effectiveness in less homophilic hypergraphs as well.

**Sensitivity of Multi-V-label Parameters.** We analyze how the parameters related to multi-v-label propagation affect the performance of VilLain, specifically the number $D$ of v-label subspaces and the number $\lceil d/D \rceil$ of v-labels in each subspace. As we can see in Figure 6, both the number of subspaces ($D$) and the number of v-labels in each subspace ($\lceil d/D \rceil$) contribute to the improvement in embedding quality. Empirically, we find that the number of v-labels per subspace has a stronger impact on the performance of VilLain.

**Additional Experimental Results.** Due to the space limit, other experimental results are provided in Appendix C including (1) usefulness as input features, (2) improvements from external node features, (3) alternative aggregation methods for embedding generation, and (4) comparisons with graph-modeling-based baselines. Furthermore, in Appendix D, we develop VilLain$_B$, a space-efficient

---

[7]We did not evaluate on Primary and High. Due to their high density, even removing 90% of their hyperedges did not result in any unobserved nodes.

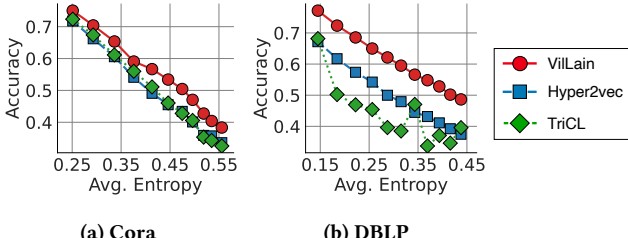

(a) Cora      (b) DBLP

**Figure 5: VilLain consistently outperforms Hyper2vec and TriCL in node classification across varying levels of average hyperedge entropy (i.e., heterophilicity).**

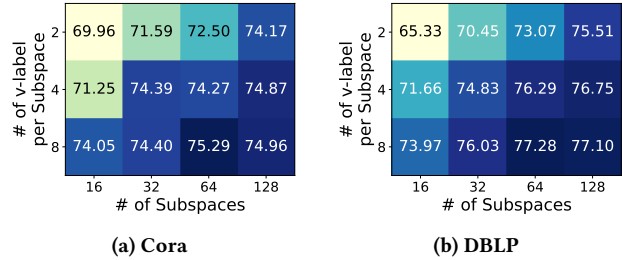

(a) Cora      (b) DBLP

**Figure 6: Both the number of subspaces ($D$) and the number of v-labels in each subspace ($\lceil d/D \rceil$) are positively correlated to the node classification accuracy.**

variant of VilLain that generates binary node embeddings for hypergraphs. Empirical results demonstrate its superior performance compared to baseline methods while requiring only 1/32 of the bits for encoding the node embedding vectors.

## 7 CONCLUSIONS, LIMITATIONS, AND FUTURE DIRECTIONS

In this work, we propose VilLain for self-supervised node representation learning on hypergraphs. VilLain learns node embeddings that reproduces higher-order label homogeneity in real-world hypergraphs, without requiring external node labels or features. We summarize our contributions as follows:

- **Empirical Findings:** We discover the higher-order homogeneity in real-world hypergraphs, which serves as a guiding principle in the design of VilLain (Section 4).
- **Algorithm Design:** We develop VilLain, a node embedding method for hypergraphs that does not require external information such as labels or features. It produces versatile embeddings that are effective for various tasks (Section 5).
- **Extensive Experiments:** We demonstrate the overall superiority of VilLain over 15 unsupervised and (semi-)supervised competitors on eight datasets in four tasks (Section 6).

While higher-order label homogeneity is observed in a majority of real-world hypergraphs, this may not hold in certain hypergraphs with heterophilic characteristics. Extending VilLain for heterophilic hypergraphs, thus, can be a promising future work.

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

# A DETAILS ON TIME/SPACE COMPLEXITY

In this section, we provide details on the time and space complexity analysis provided in Section 5.

## A.1 Details on Time Complexity

**Time complexity for v-label propagation.** Since we adopt mean-pooling aggregation of $\frac{d}{D}$ v-labels for both nodes and hyperedges in each subspace, it takes $O\left(\frac{d}{D}\sum_{e\in E}|e|\right)$. Thus, for $D$ subspaces, it takes:

$$O\left(d\sum_{e\in E}|e|\right) \text{ time} \qquad (8)$$

for each step of propagation.

**Time complexity for loss computation.** In VilLain, there are three losses, $\mathcal{L}_{\text{local}}$, $\mathcal{J}_{\text{cls}}$, and $\mathcal{J}_{\text{dst}}$ that are computed to optimize $\tilde{\mathbf{X}}$.

- For $\mathcal{L}_{\text{local}}$, the entropy of the assignment over $\frac{d}{D}$ v-labels at each node and each hyperedge at each step needs to be computed, and this takes $O\left(\frac{d}{D}(|V|+|E|)\right)$ time for each subspace. Thus, for $D$ subspaces, it takes:

$$O\left(d(|V|+|E|)\right) \text{ time} \qquad (9)$$

for each propagation step.

- For $\mathcal{J}_{\text{cls}}$, the entropy of the global assignment over $\frac{d}{D}$ v-labels needs to be computed at each step, and this takes $O\left(\frac{d}{D}(|V|+|E|)\right)$ time for each subspace. Thus, for $D$ subspaces, it takes:

$$O\left(d(|V|+|E|)\right) \text{ time} \qquad (10)$$

for each propagation step.

- For $\mathcal{J}_{\text{dst}}$, $\bar{\mathbf{x}}_1^{(\ell)}, \cdots, \bar{\mathbf{x}}_{d/D}^{(\ell)}$ and $\bar{\mathbf{y}}_1^{(\ell)}, \cdots, \bar{\mathbf{y}}_{d/D}^{(\ell)}$ are required, and it takes $O\left(\left(\frac{d}{D}\right)^2|V|\right)$ time and $O\left(\left(\frac{d}{D}\right)^2|E|\right)$ time, respectively, to compute them for each subspace. Thus, for $D$ subspaces, it takes:

$$O\left(\frac{d^2}{D}(|V|+|E|)\right) \text{ time} \qquad (11)$$

for each propagation step.

Thus, from Eq. (8)-(11), the time complexity including (a) v-label propagation and (b) loss computation is:

$$O\left(kd\sum_{e\in E}|e| + \frac{kd^2}{D}(|V|+|E|)\right).$$

**Time complexity for embedding generation.** To generate node embeddings using Eq. (3), which requires the mean-pooling propagation of v-labels for $k'$ steps, it takes:

$$O\left(k'd(|V|+|E|)\right) \text{ time.}$$

## A.2 Details on Space Complexity

**Space complexity for v-label propagation.** During its v-label propagation, VilLain stores assignment matrices of nodes $\mathbf{X}^{(\ell)}$ and hyperedges $\mathbf{Y}^{(\ell)}$ of $\frac{d}{D}$ v-labels in $D$ subspaces which requires:

$$O\left(kd(|V|+|E|)\right) \text{ space}$$

for $\ell = 1, \cdots, k$ steps.

**Space complexity for loss computation.** The losses $\mathcal{L}_{\text{local}}^{(\ell)}$ and $\mathcal{J}_{\text{cls}}^{(\ell)}$ at the $\ell^{\text{th}}$ step can be computed directly from $\mathbf{X}^{(\ell)}$ and $\mathbf{Y}^{(\ell)}$, without requiring additional storage space. On the other hand, to compute $\mathcal{J}_{\text{dst}}^{(\ell)}$ at the $\ell^{\text{th}}$ step, $\bar{\mathbf{x}}_1^{(\ell)}$ and $\bar{\mathbf{y}}_1^{(\ell)}$ are used, which are computed based on the pairwise cosine similarity between $d/D$ v-labels, requiring $O\left(\frac{d^2}{D}(|V|+|E|)\right)$ space for $D$ subspaces. Thus, the total space required for $\ell = 1, \cdots, k$ steps is;

$$O\left(\frac{kd^2}{D}(|V|+|E|)\right).$$

Note that unlike GNN-based methods [17, 20], VilLain does not have any additional learnable parameters in each layer.

**Space complexity for embedding generation.** To generate node embeddings, $\mathbf{X}^{(\ell)}$ and hyperedge embeddings $\mathbf{Y}^{(\ell)}$ for $\ell = 1, \cdots, k'$ steps are used, and thus $O\left(k'd(|V| + |E|)\right)$ space is required.

## B  DETAILS ON EXPERIMENTAL SETTINGS

Here, we provide detailed information on experimental settings.

### B.1  Details of Datasets

The statistics of the datasets we used are shown in Table 1.

**Preprocessing** For all datasets, we use the largest connected component of the original hypergraph. We process the huge Amazon by remaining nodes that are from the 10 most frequently appeared labels. Then, we randomly sample 1% of the nodes from each label.

**Ground-truth labels** Here, we provide how the ground-truth labels of each dataset are assigned. In Primary and High, each node is a person (e.g., student or teacher), and each hyperedge indicate a group interaction among them. If a person is a teacher, then he or she is labeled as a teacher. Otherwise, students are labeled based on the classroom they belong to. In Citeseer, Cora, and Pubmed, which are co-citation hypergraphs, each node is a paper and each hyperedge is a paper that cited the paper. In these hypergraphs, nodes are assigned by their categories. In DBLP, which is a collaboration hypergraph, each node is a paper and each hyperedge is the set of papers written by the same author. Nodes are labeled by their categories. In Trivago, each node is a hotel and each hyperedge is a set of hotels that were clicked in a Web browsing session. Each node is labeled by the location, specifically, the country where the hotel is located. In Amazon, each node is a product, and each hyperedge is a set of products that were co-purchased. Labels of the nodes are assigned by the product categories.

### B.2  Baselines & Hyperparameters

In this subsection, we discuss the hyperparameters that are used for each method. The implementations we used to run baseline methods are listed in Table 7. Since we consider the unsupervised setting, specifically, without using any labels, the models used for evaluation should be selected without validating on hold-out labeled data. Thus, for unsupervised baseline methods, we either used their default hyperparameter settings or try to find the settings that generally work well across all datasets. However, for (semi-)supervised methods, we use the validation set to tune their hyperparameters.

In VilLain, we fix the number of propagation steps for training to $k = 4$, and for inference, we use $k' = 10$ for small datasets (i.e., Primary, High, Cora, Citeseer, Pubmed) and $k' = 100$ for large datasets (i.e., DBLP, Amazon, and Trivago). The learning rate is fixed to 0.01, and the explained variance ratio of the PCA used in VilLain is fixed to 0.99, throughout the experiments.

For Deepwalk [51] and Node2vec [23], we use the default hyperparameters. Specifically, we set the number of walks to 10, the length of each walk to 80, the window size to 5, and the learning rate to 0.05. For $p$ and $q$ in Node2vec, we use 1 for both.

For DGI [61], we use the PReLu for the activation function and set the learning rate to 0.001, as given as default.

For GRACE [85], we use the ReLU for the activation function and the number of GCN layers is set to 2. The learning rate and

**Table 7: Open source links to the baseline source codes.**

| Method | Github Link |
|---|---|
| GCN | https://pytorch-geometric.readthedocs.io |
| GAT | https://pytorch-geometric.readthedocs.io |
| Deepwalk | https://github.com/benedekrozemberczki/karateclub |
| Node2vec | https://github.com/benedekrozemberczki/karateclub |
| DGI | https://github.com/PetarV-/DGI |
| GRACE | https://github.com/CRIPAC-DIG/GRACE |
| GMI | https://github.com/zpeng27/GMI |
| HGNN | https://github.com/iMoonLab/HGNN |
| HNHN | https://github.com/twistedcubic/HNHN |
| AllSet | https://github.com/jianhao2016/AllSet |
| UniGNN | https://github.com/OneForward/UniGNN |
| HyperGCL | https://github.com/weitianxin/HyperGCL |
| Hyper2vec | https://github.com/jeffhj/NHNE |
| TriCL | https://github.com/wooner49/TriCL |

the weight decay rate are set to 0.001 and 0.00001, respectively. Regarding augmentations (e.g., edge drop and feature drop), all rates are set to 0.2. The dimension of the projection head is set to be the same as the hidden dimension.

For GMI [50], we use the PReLU for the activation function. The learning rate is set to 0.001 without weight decaying. There are three additional hyperparameters $\alpha$, $\beta$, and $\gamma$ that determine the weights of the local and global mutual information, and they are set to $\alpha = 0.8$, $\beta = 1.0$, and $\gamma = 1.0$, as the default values provided by the authors.

For HyperGCL [64], we use their default hyperparameters. The number of epochs is set to 500, the augmentation ratio is set to 0.3, the temperature is set to 0.3, and the dropout rate is set to 0.2.

For Hyper2vec [27], we use their default hyperparameters. The number of walks is set to 10 and the length of each walk is set to 20. The size of the window is 5 and two additional parameters $p$ and $q$ are both set to 1.

For LBSN [73], the number of negative samples and the learning rate are set to 10 and 0.01, respectively.

For TriCL [35], we set the number of GCN layers to 1 since it was given as default hyperparameters for most datasets. The learning rate and the weight decaying rate are set to 0.0005 and 0.00001, respectively. Regarding the data augmentation, the drop rates for node features and the incidence matrix are both set to 0.4. Three temperature hyperparameters, $\tau_n$, $\tau_g$, and $\tau_m$ are all set to 0.5, and two weight hyperparameters $w_g$ and $w_m$ are set to 4 and 1, respectively.

### B.3  Node Retrieval Protocol

To perform the node retrieval task, we sample $\min(|V|, 1000)$ query nodes from the hypergraph uniformly at random. For each query node, we rank the nodes, excluding the query node, based on the cosine similarity between their learned embeddings and the that of the query node. Then, we measure the Mean Average Precision (MAP), which is commonly employed in information retrieval tasks (e.g., computer vision [8, 38] or natural language processing [3, 59]). Here, we define nodes with labels same as that of the query node as the ground-truth. Thus, the MAP yields a higher score when nodes belonging to the same class as the query node are ranked highly.

**Table 8: The number of label propagation steps required for the average entropy of the hyperedges in the real-world hypergraphs to reach $\epsilon$ of that of the hyperedges in the randomized hypergraphs.**

|  | PR | HG | CS | CR | PM | DB | TV | AZ |
|---|---|---|---|---|---|---|---|---|
| $\epsilon = 0.9$ | 6 | 15 | 140 | 31 | 16 | 832 | 812 | 22 |
| $\epsilon = 0.99$ | 15 | 34 | 456 | 102 | 43 | 2,813 | 2,234 | 47 |
| $\epsilon = 0.999$ | 22 | 47 | $\infty$ | 161 | 70 | 4,046 | 3,409 | 59 |

## B.4 Hyperedge Prediction Protocol

To perform the hyperedge prediction task, we first split the original hypergraph $G = (V, E)$ into two sub-hypergraphs $G_{\text{train}} = (V_{\text{train}}, E_{\text{train}})$ and $G_{\text{test}} = (V_{\text{test}}, E_{\text{test}})$ where $E = E_{\text{train}} \cup E_{\text{test}}$ and $E_{\text{train}} \cap E_{\text{test}} = \varnothing$. We also ensure that all nodes are contained in $G_{\text{train}}$ (i.e., $V_{\text{train}} = V$) so that embeddings of all nodes in $G$ are learned. Given a train ratio $\gamma$, we set the number of hyperedges in $G_{\text{train}}$ and $G_{\text{test}}$ to be divided based on it, i.e., $|E_{\text{train}}| : |E_{\text{test}}| = \gamma : 1 - \gamma$. Specifically, we set $\gamma = 0.80$ for all datasets except for Amazon, which is relatively very sparse, and thus we set $\gamma = 0.95$.

Once we obtain node embeddings of all nodes $V$, we generate sets of fake hyperedges $E_{\text{train}}^{\text{fake}}$ and $E_{\text{test}}^{\text{fake}}$ as counterparts of true hyperedges $E_{\text{train}}$ and $E_{\text{test}}$. Specifically, for each true hyperedge $e \in E_{\text{train}}$ (or $E_{\text{test}}$), we randomly sample $|e|$ nodes from $V$ and create $e' \in E_{\text{train}}^{\text{fake}}$ (or $E_{\text{test}}^{\text{fake}}$). Then, a logistic regression classifier is trained on the $E_{\text{train}} \cup E_{\text{train}}^{\text{fake}}$ and the performance of the hyperedge prediction is evaluated on $E_{\text{test}} \cup E_{\text{test}}^{\text{fake}}$.

## C ADDITIONAL EXPERIMENTAL RESULTS

In this section, we provide additional experimental results that are not covered in the main context.

## C.1 Higher-Order Homogeneity

We examine the number of steps of label propagation required for the average entropy of the hyperedges in the real-world hypergraphs to reach $\epsilon$ of that of the hyperedges in the randomized hypergraphs. For example, it requires 3,409 steps of label propagation to reach 0.999 of the average entropy of random hypergraphs, as shown in Table 8. These results support Observation 2, i.e., real-world hypergraphs exhibit not only the hyperedge-level label homogeneity but also the higher-order homogeneity.

## C.2 Full Results

We provide the full results on the three considered downstream tasks: node classification (Table 16), hyperedge prediction (Table 17), node clustering (Table 18), and node retrieval (Table 19). In these tables, we include the results of the space-efficient version, VilLain$_B$ (see Appendix D). For VilLain$_B$, we consider two variants, VilLain$_B^{128}$ and VilLain$_B^{256}$, which generate binary embeddings that cost 128 and 256 bits, respectively, for each embedding vector. We set the number of v-labels in each subspace to 4 for both variants. Note that VilLain$_B^{128}$ and VilLain$_B^{256}$ require only **1/32** and **1/16** of the space used by the other methods, respectively. We include the standard deviation in the tables. In node classification, hyperedge prediction, and node retrieval tasks, on average,

**Table 9: Density (i.e., $|E|/|V|$) and overlapness (i.e., $\sum_{e \in E} |e|/|V|$) of each dataset. Primary exhibits exceptionally high density and overlapness compared to other datasets.**

|  | PR | HG | CS | CR | PM | DB | TV | AZ |
|---|---|---|---|---|---|---|---|---|
| Density | 52.495 | 23.908 | 0.803 | 1.130 | 2.079 | 0.522 | 1.350 | 0.122 |
| Overlapness | 126.979 | 55.633 | 2.755 | 3.457 | 9.049 | 2.510 | 4.207 | 1.622 |

**Table 10: VilLain benefits from input node features in node classification. When utilizing node features, it ranks highest on average among its feature-requiring baselines across four datasets where node features are provided.**

| Method | DBLP | Citeseer | Cora | Pubmed | Rank |
|---|---|---|---|---|---|
| GCN | 84.45 ± 1.25 | 64.60 ± 3.00 | 76.06 ± 2.29 | 74.92 ± 2.90 | 5.25 ± 2.62 |
| GAT | 77.07 ± 1.63 | 50.39 ± 3.40 | 59.79 ± 2.08 | 73.96 ± 2.13 | 11.00 ± 1.15 |
| DGI | 85.64 ± 1.13 | 68.53 ± 2.91 | 77.50 ± 2.04 | 75.62 ± 2.82 | 3.50 ± 2.38 |
| GRACE | 85.63 ± 1.05 | 61.33 ± 2.78 | 71.16 ± 1.81 | 77.47 ± 1.57 | 5.75 ± 3.30 |
| GMI | 80.85 ± 1.49 | 57.09 ± 2.68 | 74.73 ± 1.69 | 76.38 ± 2.21 | 8.00 ± 2.44 |
| HGNN | 84.36 ± 1.70 | 64.28 ± 2.53 | 75.63 ± 1.39 | 76.63 ± 2.44 | 5.00 ± 0.81 |
| HNHN | 74.44 ± 1.98 | 58.53 ± 3.31 | 67.87 ± 3.51 | 69.38 ± 3.47 | 10.75 ± 1.25 |
| AllSet | 83.67 ± 1.53 | 57.88 ± 3.14 | 70.07 ± 3.23 | 75.24 ± 2.93 | 9.00 ± 1.15 |
| UniGNN | 84.22 ± 1.57 | 63.79 ± 3.72 | 74.44 ± 2.50 | 76.99 ± 2.82 | 5.75 ± 1.89 |
| HyperGCL | 76.12 ± 6.04 | 63.30 ± 2.11 | 73.01 ± 3.68 | **82.62 ± 3.25** | 6.75 ± 4.19 |
| TriCL | **86.59 ± 0.88** | 64.53 ± 3.17 | **79.03 ± 0.63** | 76.60 ± 1.71 | 2.75 ± 2.06 |
| VilLain | 85.68 ± 0.85 | **68.77 ± 1.82** | 76.54 ± 1.44 | 78.25 ± 2.41 | **2.00 ± 0.81** |

VilLain and VilLain$_B$ show the best performance. In the node clustering task, VilLain show the second-best performance. Notably, VilLain$_B^{128}$ and VilLain$_B^{256}$, which require substantially less number of bits for embeddings than the other, highly rank on average. Moreover, it is worthwhile to notice that the proposed methods outperform (semi-)supervised methods (e.g., HGNN and AllSet), which are trained specifically for the node classification task. We conjecture that v-label propagation inherits rich structural properties and also potential higher-order structure-label relationships, generating high-quality representations of nodes.

## C.3 When $\mathcal{L}_{\text{global}}$ is Important

As shown in Table 4 in Section 6.3, VilLain outperforms VilLain-L in most datasets. Notably, this performance advantage is particularly significant in Primary, and in this subsection, we analyze the reasons behind this improvement and explore when the inclusion of $\mathcal{L}_{\text{global}}$ is particularly beneficial. We hypothesize that VilLain-L faces difficulty in learning distinctive v-label distributions, with a single v-label accounting for nearly 100% of nodes in Primary, regardless of the predefined number of v-labels. This challenge may arise due to the dataset's unique characteristic of densely connected nodes. This is supported by the measured density (i.e., $|E|/|V|$) and overlapness (i.e., $\sum_{e \in E} |e|/|V|$) of the hypergraphs in Table 9.

## C.4 Improvements from Node Features

External node features, if available, are useful and typically enhance method performance. VilLain can be extended to incorporate node features by introducing $|V|$ additional hyperedges, where each hyperedge is a group of the $k$-nearest neighbors of each node based on cosine similarity between node features. Then, it learns v-label

**Table 11: The average accuracy over all feature-requiring methods (e.g., GCN, HGNN, and TriCL) using different input features. Hyper2vec is the most useful input feature, compared to learnable embeddings and Node2vec.**

| Method | DBLP | Trivago | Amazon | Primary | High | Citeseer | Cora | Pubmed | Rank |
|---|---|---|---|---|---|---|---|---|---|
| Learnable | $21.87_{\pm 3.72}$ | $6.75_{\pm 6.64}$ | $\underline{13.88}_{\pm 5.73}$ | $61.71_{\pm 24.20}$ | $74.19_{\pm 31.24}$ | $41.73_{\pm 9.26}$ | $44.80_{\pm 13.02}$ | $55.11_{\pm 11.28}$ | $2.87_{\pm 0.33}$ |
| Node2vec | $\underline{36.21}_{\pm 5.07}$ | $\underline{25.11}_{\pm 11.47}$ | $\mathbf{26.63}_{\pm 5.75}$ | $\mathbf{81.87}_{\pm 10.21}$ | $\mathbf{95.97}_{\pm 4.92}$ | $\underline{53.37}_{\pm 4.18}$ | $\underline{54.81}_{\pm 6.48}$ | $\underline{71.32}_{\pm 5.34}$ | $\underline{1.62}_{\pm 0.48}$ |
| Hyper2vec | $\mathbf{63.63}_{\pm 6.06}$ | $\mathbf{55.51}_{\pm 23.03}$ | OOT | $\underline{81.79}_{\pm 11.54}$ | $\underline{92.60}_{\pm 6.48}$ | $\mathbf{57.94}_{\pm 3.27}$ | $\mathbf{70.05}_{\pm 3.93}$ | $\mathbf{74.60}_{\pm 4.85}$ | $\mathbf{1.28}_{\pm 0.45}$ |

**Table 12: HGNN and TriCL yield unsatisfactory performance with learnable features and random features, while input features learned by Hyper2vec demonstrate significantly better accuracy. VilLain outperforms them by a large margin.**

| Method | | DBLP | Primary | High | Citeseer | Cora | Pubmed | Rank |
|---|---|---|---|---|---|---|---|---|
| HGNN | Learnable Features | $21.47_{\pm 2.28}$ | $76.71_{\pm 3.36}$ | $79.58_{\pm 3.48}$ | $42.34_{\pm 2.11}$ | $43.02_{\pm 2.33}$ | $54.66_{\pm 2.76}$ | $4.67_{\pm 0.47}$ |
| | Random Features | $21.24_{\pm 1.91}$ | $78.43_{\pm 2.36}$ | $83.12_{\pm 3.65}$ | $43.04_{\pm 3.39}$ | $43.26_{\pm 2.44}$ | $54.41_{\pm 3.46}$ | $4.33_{\pm 0.47}$ |
| | Hyper2vec | $66.60_{\pm 2.18}$ | $88.28_{\pm 5.02}$ | $92.19_{\pm 3.84}$ | $60.91_{\pm 2.32}$ | $72.90_{\pm 2.00}$ | $76.58_{\pm 2.86}$ | $2.67_{\pm 0.45}$ |
| TriCL | Learnable Features | $19.31_{\pm 1.11}$ | $31.86_{\pm 2.64}$ | $30.34_{\pm 3.75}$ | $24.94_{\pm 1.62}$ | $25.10_{\pm 2.22}$ | $38.74_{\pm 2.25}$ | $6.50_{\pm 0.50}$ |
| | Random Features | $18.96_{\pm 1.20}$ | $31.84_{\pm 3.49}$ | $38.33_{\pm 4.42}$ | $25.89_{\pm 2.28}$ | $24.29_{\pm 1.74}$ | $39.69_{\pm 1.93}$ | $6.50_{\pm 0.50}$ |
| | Hyper2vec | $68.18_{\pm 1.36}$ | $92.67_{\pm 2.50}$ | $98.10_{\pm 1.02}$ | $59.17_{\pm 3.35}$ | $72.35_{\pm 1.53}$ | $78.57_{\pm 1.88}$ | $\underline{2.33}_{\pm 0.45}$ |
| VilLain | | $\mathbf{77.16}_{\pm 1.26}$ | $\mathbf{93.66}_{\pm 3.93}$ | $\mathbf{99.19}_{\pm 0.41}$ | $\mathbf{61.53}_{\pm 3.17}$ | $\mathbf{75.03}_{\pm 1.38}$ | $\mathbf{78.82}_{\pm 1.47}$ | $\mathbf{1.00}_{\pm 0.00}$ |

distributions on an augmented hypergraph with $|V|$ nodes and $|V| + |E|$ hyperedges. As shown in Table 10, VilLain benefits from using node features, outperforming its feature-requiring baselines in terms of average ranks when using $k = 3$.

We would like to emphasize that our simple approach to utilizing external node features is distinguished from how other baseline methods utilize them (i.e., projecting and propagating them through edges), potentially making it a suboptimal choice. However, it is crucial to note that VilLain is primarily designed for scenarios where node features are unavailable and thus is tailored to perform best in such cases. Furthermore, it is important to note that in our other experiments, we used topological node features obtained through Hyper2vec, instead of external features for the baselines that require input node features.

### C.5 Usefulness as Input Features

We evaluate the usefulness of the methods as an input of the feature-requiring methods (i.e., GCN, GAT, DGI, GRACE, GMI, HGNN, HNHN, AllSet, UniGNN, HyperGCL, and TriCL). Specifically, we train these models using three different input features including a learnable one, which is trained together with the models. As shown in Table 11, using Hyper2vec yields the best accuracy in node classification, and thus we use their embeddings for input features of feature-requiring methods.

In addition, in Table 12, we present a comparison of node classification accuracies of HGNN and TriCL, which are semi-supervised and self-supervised GNN methods for hypergraphs, respectively. We utilize different input features across the considered datasets, except for those that result in out-of-memory issues. We can see that GNNs with learnable features and random features yield unsatisfactory performance, while input features learned by Hyper2vec demonstrate significantly better accuracy. Most importantly, VilLain outperforms them by a large margin.

### C.6 Effects of Long-Range V-label Propagation

To examine the effects of the long-range propagation of v-labels, we test how the number of steps $k$ (during training) and $k'$ (during inference) affect the performance of VilLain in the three considered tasks in Tables 20 and 21, respectively. Except for Primary and High, which are the smallest datasets, adopting long-range propagation of v-labels is beneficial for node classification, node retrieval, and hyperedge prediction. In particular, we can see that large datasets (e.g., DBLP, Trivago, and Amazon) benefit from large $k$s and $k'$s.

### C.7 Aggregation Method for Embedding Generation

As discussed in Section 5.1, we aggregate embeddings obtained with various numbers of v-labels. While the aggregation method is flexible, we concatenate embeddings obtained using different numbers of v-labels, specifically, for each number $\lceil \frac{d}{D} \rceil \in \{2, 3, \cdots, 8\}$ of v-labels, we learn $D$ subspaces and then perform PCA to ensure that the final embedding is of the target dimension $d$. In Table 13, we compare the performance with VilLain when applying mean-pooling, instead of PCA, to the embeddings from different v-label numbers, for the embedding aggregation. Across three different downstream tasks, the concatenate-then-PCA outperforms mean-pooling on average.

### C.8 Hyperedge Embedding Method for Hyperedge Prediction

To obtain the embedding of each hyperedge, we apply maxmin pooling, i.e., elementwise max pooling - elementwise min pooling, to the embeddings of the nodes in it. In Table 14, we test the effectiveness of maxmin pooling compared to mean pooling for hyperedge prediction. For both VilLain and TriCL [35], which is the strongest baseline, maxmin pooling is more effective than mean pooling across all datasets.

**Table 13: To aggregate embeddings obtained from various numbers of v-labels in each subspace, concatenating the embeddings and applying PCA (PCA) outperforms averaging the embeddings (mean) in the three considered downstream tasks.**

| | Method | DBLP | Trivago | Amazon | Primary | High | Citeseer | Cora | Pubmed | Rank |
|---|---|---|---|---|---|---|---|---|---|---|
| NCS | Mean | 74.56 ± 1.14 | 77.23 ± 1.35 | 56.36 ± 2.23 | **93.88** ± 3.94 | 98.95 ± 0.70 | **62.70** ± 2.78 | 74.38 ± 1.31 | **79.03** ± 1.64 | 1.62 ± 0.48 |
| | PCA | **77.16** ± 1.26 | **79.43** ± 1.63 | **57.95** ± 2.47 | 93.66 ± 3.93 | **99.19** ± 0.41 | 61.53 ± 3.17 | **75.03** ± 1.38 | 78.82 ± 1.47 | **1.37** ± 0.48 |
| HP | Mean | 80.37 ± 0.97 | 95.11 ± 0.55 | 94.81 ± 0.37 | 82.40 ± 0.89 | 87.21 ± 0.67 | **82.66** ± 0.95 | **79.44** ± 0.57 | **83.10** ± 0.70 | 1.62 ± 0.48 |
| | PCA | **81.61** ± 0.52 | **95.12** ± 0.37 | **94.91** ± 0.36 | **83.19** ± 0.56 | **87.79** ± 0.68 | 82.08 ± 1.42 | 78.95 ± 0.79 | 82.79 ± 0.79 | **1.37** ± 0.48 |
| NCT | Mean | 46.32 ± 1.36 | 65.77 ± 0.32 | 34.77 ± 0.50 | **85.90** ± 1.30 | 98.72 ± 0.00 | 34.04 ± 0.86 | 48.38 ± 0.95 | 32.62 ± 0.02 | 1.75 ± 0.43 |
| | PCA | **46.58** ± 0.62 | **69.35** ± 0.32 | **35.24** ± 0.48 | 85.67 ± 1.88 | 98.72 ± 0.00 | **34.53** ± 0.45 | **50.38** ± 2.25 | **32.73** ± 0.00 | **1.12** ± 0.22 |
| NR | Mean | 49.35 ± 0.00 | 43.84 ± 0.00 | 51.26 ± 0.72 | 86.68 ± 0.00 | 98.78 ± 0.00 | 43.95 ± 0.10 | 52.76 ± 0.30 | 63.12 ± 0.37 | 2.00 ± 0.00 |
| | PCA | **60.15** ± 0.55 | **67.23** ± 0.72 | **53.64** ± 0.47 | **91.26** ± 0.00 | **98.99** ± 0.00 | **46.37** ± 0.00 | **57.96** ± 0.07 | **64.43** ± 0.07 | **1.00** ± 0.00 |

**Table 14: To obtain the embedding of each hyperedge, maxmin-pooling is more effective than mean-pooling in all datasets in both VilLain and TriCL (the strongest considered baseline method).**

| | Method | DBLP | Trivago | Amazon | Primary | High | Citeseer | Cora | Pubmed | Rank |
|---|---|---|---|---|---|---|---|---|---|---|
| VilLain | Mean | 52.55 ± 1.13 | 59.36 ± 1.52 | 64.99 ± 2.34 | 57.54 ± 1.79 | 56.83 ± 1.81 | 54.03 ± 2.20 | 56.89 ± 2.38 | 57.56 ± 0.99 | 2.00 ± 0.00 |
| | Maxmin | **81.61** ± 0.52 | **95.12** ± 0.37 | **94.91** ± 0.36 | **83.19** ± 0.56 | **87.79** ± 0.68 | **82.08** ± 1.42 | **78.95** ± 0.79 | **82.79** ± 0.79 | **1.00** ± 0.00 |
| TriCL | Mean | 53.60 ± 0.89 | OOM | OOM | 63.84 ± 0.97 | 59.29 ± 1.08 | 61.33 ± 2.40 | 68.28 ± 1.50 | 59.69 ± 1.34 | 2.00 ± 0.00 |
| | Maxmin | **77.40** ± 0.68 | OOM | OOM | **83.00** ± 0.63 | **87.78** ± 0.52 | **81.96** ± 0.91 | **76.69** ± 0.70 | **80.45** ± 0.75 | **1.00** ± 0.00 |

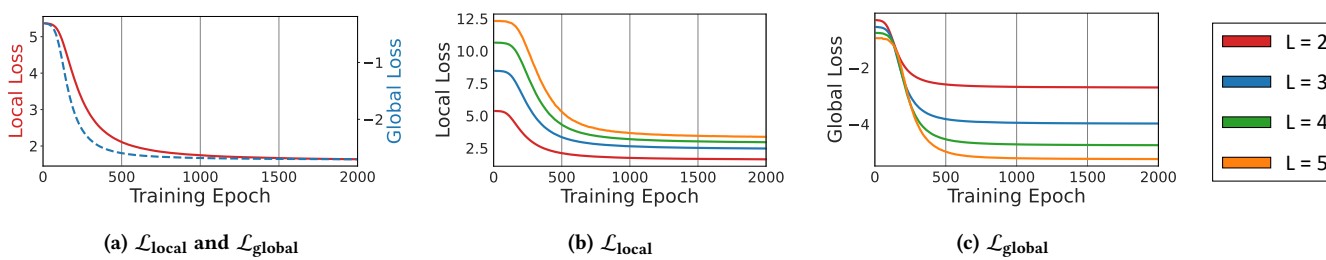

(a) $\mathcal{L}_{\text{local}}$ and $\mathcal{L}_{\text{global}}$      (b) $\mathcal{L}_{\text{local}}$      (c) $\mathcal{L}_{\text{global}}$

**Figure 7: Losses with respect to training epochs in VilLain. (a) The two losses $\mathcal{L}_{\text{local}}$ and $\mathcal{L}_{\text{global}}$ are jointly optimized in VilLain. (b) Optimization with smaller numbers of v-labels is easier to optimize $\mathcal{L}_{\text{local}}$. (c) On the other hand, optimization with larger numbers of v-labels is easier to optimize $\mathcal{L}_{\text{global}}$.**

**Table 15: VilLain, applied to hypergraphs, outperforms the recent graph-based baseline methods (i.e., GATv2, GCNII, and GPRGNN) across benchmark graph datasets.**

| Graph Type | Model | Cora | Citeseer | Pubmed | Avg. Rank |
|---|---|---|---|---|---|
| Graph | GATv2 | 70.58 ± 1.4 | 52.76 ± 2.4 | 72.92 ± 2.9 | 3.33 ± 0.47 |
| | GCNII | 71.37 ± 2.4 | 56.47 ± 1.6 | 74.63 ± 2.0 | 2.00 ± 0.00 |
| | GPRGNN | 69.60 ± 2.0 | 56.38 ± 2.2 | 71.44 ± 1.6 | 3.66 ± 0.47 |
| Hypergraph | VilLain | **75.03** ± 1.38 | **61.53** ± 3.17 | **78.82** ± 1.47 | **1.00** ± 0.00 |

## C.9 Comparison with Graph-Modeling-Based Methods

In Section 6, we applied GNNs to pairwise graphs which are transformed from hypergraphs. For this transformation, we adopted clique expansion, which is a popular approach to transform hypergraphs into graphs [72, 83, 84]. However, such clique-expanded graphs are often different from graphs conventionally used for GNN benchmarks. Specifically, for the citation datasets (e.g., Cora, Pubmed, and Citeseer), each edge joins co-cited graphs in clique-expanded graphs, while each edge in GNN-benchmark graphs joints

a pair of citing and cited papers. Thus, we evaluate the performance of well-established GNN models, specifically GATv2 [5], GCNII [9], and GPRGNN [12], when applied to the original structures of the graph datasets. We consider the setting without features, which our paper focuses on and thus use embeddings obtained from Node2vec as their input features. As shown in Table 15, VilLain outperforms GNN competitors in node classification, even when they use graphs modeled with the same semantics as hypergraphs, instead of clique expansion. This demonstrates the effectiveness of employing hypergraph modeling and VilLain for learning embeddings from its structure.

## C.10 Loss of VilLain

We examine how losses of VilLain decrease with training epochs. As discussed in Section 5, VilLain optimizes two losses $\mathcal{L}_{\text{local}}$ and $\mathcal{L}_{\text{global}}$ that aim to capture the local and global structural information of the hypergraph, respectively. As shown in Figure 7a, the two losses $\mathcal{L}_{\text{local}}$ and $\mathcal{L}_{\text{global}}$ jointly decrease as VilLain is trained. In terms of the number $L = \lceil d/D \rceil$ of v-labels in each subspace, the decrease of $\mathcal{L}_{\text{local}}$ is facilitated by smaller $L$. Intuitively, a smaller number of v-labels is more likely to lead to homogeneous hyperedges. On the other hand, $\mathcal{L}_{\text{global}}$ decreases faster with a larger number of v-labels in each subspace since more diverse v-labels are more likely to be distinctive from each other.

# D VILLAIN$_{\text{B}}$: SPACE-EFFICIENT BINARY EMBEDDING

As hypergraphs grow in size, so does the space required to store the embeddings. Specifically, a continuous $d$-dimensional vector consisting of $d$ real numbers requires $32d$ bits if float-32 is used to represent each real number. To reduce the space requirement, we propose VilLain$_{\text{B}}$, a space-efficient version of VilLain that produces binary node embeddings for hypergraphs. Specifically, we binarize the continuous vector $\mathbf{Z}_i^{\langle t \rangle}$ of node $v_i$ in each $t^{\text{th}}$ subspace obtained by VilLain, which is a probabilistic distribution over $d/D$ v-labels, to a one-hot vector $\widehat{\mathbf{Z}}_i^{\langle t \rangle} \in \{0, 1\}^{d/D}$ as:

$$\widehat{\mathbf{Z}}_i^{\langle t \rangle} = \text{one-hot}\left(\arg\max_j \left(\mathbf{Z}_{i,j}^{\langle t \rangle}\right)\right).$$

Then, the final binarized embedding $\widehat{\mathbf{Z}}_i \in \{0, 1\}^d$ is obtained by concatenating the binarized embeddings from the $D$ subspaces.

To encode a $d/D$-dimensional one-hot vector in each subspace, $\lceil \log_2 \frac{d}{D} \rceil$ bits are required. Hence, encoding a final binarized vector, which is the concatenation of $D$ one-hot vectors, requires $\lceil D \log_2 \frac{d}{D} \rceil$ bits. Note that, $D \log_2 \frac{d}{D} < 32d$ always holds for any positive integers $d$ and $D$ ($\leq d$).

In Tables 16, 17, 18, and 19, we include the performance of VilLain$_{\text{B}}$. While requiring a substantially smaller number of bits to encode embeddings, VilLain$_{\text{B}}$ outperforms baseline methods in the considered four downstream tasks.

**Table 16: Full results on node classification (in terms of accuracy). VilLain and VilLain$_B$ outperform the existing (hyper)graph representation learning methods.**

| Method | DBLP | Trivago | Amazon | Primary | High | Citeseer | Cora | Pubmed | Rank |
|---|---|---|---|---|---|---|---|---|---|
| GCN | 49.65 ± 2.91 | 18.53 ± 3.61 | 19.08 ± 2.43 | 89.54 ± 2.49 | 89.82 ± 4.99 | 53.35 ± 3.76 | 63.71 ± 2.73 | 70.89 ± 1.60 | 12.62 ± 2.64 |
| GAT | OOM | OOM | OOM | 58.48 ± 7.12 | 76.94 ± 9.60 | 51.06 ± 4.29 | 62.74 ± 3.07 | 61.66 ± 5.27 | 16.60 ± 1.35 |
| Deepwalk | 29.03 ± 1.43 | 16.85 ± 0.45 | 25.43 ± 1.72 | 84.89 ± 3.67 | 99.31 ± 0.48 | 45.10 ± 3.18 | 56.58 ± 1.88 | 68.58 ± 2.60 | 13.00 ± 5.04 |
| Node2vec | 29.21 ± 1.89 | 16.88 ± 0.44 | 25.27 ± 2.36 | 83.53 ± 3.09 | 99.38 ± 0.45 | 45.37 ± 3.17 | 59.15 ± 1.84 | 69.05 ± 3.00 | 12.37 ± 5.09 |
| DGI | 62.37 ± 3.32 | 73.46 ± 1.22 | 31.80 ± 1.45 | 86.66 ± 4.51 | 92.49 ± 0.60 | 61.36 ± 2.91 | 71.23 ± 2.04 | 77.51 ± 1.38 | 8.37 ± 3.87 |
| GRACE | 71.86 ± 2.51 | OOM | OOM | 63.78 ± 5.12 | 99.03 ± 0.30 | 61.16 ± 2.78 | 73.43 ± 1.81 | 77.70 ± 1.81 | 6.50 ± 4.85 |
| GMI | 64.19 ± 1.63 | OOM | OOM | 80.10 ± 4.94 | 96.61 ± 2.63 | 58.67 ± 2.68 | 71.31 ± 1.69 | 75.51 ± 2.77 | 10.66 ± 2.05 |
| HGNN | 66.60 ± 2.18 | OOM | OOM | 88.28 ± 5.02 | 92.19 ± 3.84 | 60.91 ± 2.32 | 72.90 ± 2.00 | 76.58 ± 2.86 | 8.66 ± 3.19 |
| HNHN | 63.99 ± 2.21 | 59.52 ± 1.64 | 28.99 ± 2.63 | 91.31 ± 2.47 | 96.83 ± 1.25 | 59.02 ± 1.63 | 68.81 ± 1.26 | 75.33 ± 1.77 | 8.87 ± 2.08 |
| AllSet | 63.67 ± 1.89 | 36.58 ± 0.93 | 21.75 ± 1.67 | 85.94 ± 3.02 | 95.70 ± 1.66 | 56.08 ± 1.95 | 67.73 ± 1.81 | 74.11 ± 2.04 | 11.75 ± 1.19 |
| UniGNN | 67.16 ± 2.15 | 69.98 ± 1.60 | 33.77 ± 3.22 | 88.88 ± 3.58 | 95.12 ± 3.97 | 59.10 ± 2.76 | 71.44 ± 1.03 | 74.37 ± 2.10 | 8.37 ± 2.91 |
| HyperGCL | 58.72 ± 1.54 | 74.99 ± 1.23 | 22.86 ± 2.01 | 74.07 ± 6.06 | 85.79 ± 8.92 | 57.54 ± 1.61 | 74.99 ± 1.33 | 78.44 ± 3.33 | 9.37 ± 5.67 |
| Hyper2vec | 67.18 ± 1.78 | 75.82 ± 1.45 | OOT | 92.52 ± 2.45 | 96.34 ± 1.34 | 61.50 ± 2.60 | 71.79 ± 1.63 | 77.04 ± 1.51 | 5.85 ± 2.84 |
| LBSN | 22.63 ± 2.20 | 47.99 ± 0.82 | 11.56 ± 0.90 | 86.71 ± 3.71 | 95.87 ± 2.28 | 45.43 ± 2.15 | 59.70 ± 1.31 | 54.89 ± 2.38 | 13.62 ± 3.27 |
| TriCL | 68.18 ± 1.36 | OOM | OOM | 92.67 ± 2.50 | 98.10 ± 1.02 | 59.17 ± 3.35 | 72.35 ± 1.53 | 78.57 ± 1.88 | 5.44 ± 2.13 |
| **VilLain$_B^{128}$** | 67.99 ± 1.16 | 64.93 ± 1.76 | 52.37 ± 1.82 | **95.63 ± 0.28** | 99.32 ± 0.17 | 60.83 ± 2.82 | 74.40 ± 1.38 | 77.57 ± 1.61 | 4.25 ± 1.98 |
| **VilLain$_B^{256}$** | 70.39 ± 1.76 | 69.26 ± 1.45 | 52.40 ± 2.03 | 95.15 ± 2.04 | 99.14 ± 0.29 | 60.27 ± 2.97 | 74.46 ± 1.88 | 78.00 ± 1.20 | 4.00 ± 1.73 |
| **VilLain** | **77.16 ± 1.26** | **79.43 ± 1.63** | **57.95 ± 2.47** | 93.66 ± 3.93 | 99.19 ± 0.41 | **61.53 ± 3.17** | **75.03 ± 1.38** | **78.82 ± 1.47** | **1.62 ± 1.11** |

**Table 17: Full results on hyperedge prediction (in terms of accuracy). VilLain and VilLain$_B$ (see Appendix D) outperform the existing (hyper)graph representation learning methods.**

| Method | DBLP | Trivago | Amazon | Primary | High | Citeseer | Cora | Pubmed | Rank |
|---|---|---|---|---|---|---|---|---|---|
| Deepwalk | 63.90 ± 0.94 | 61.27 ± 1.14 | 69.36 ± 0.74 | 83.79 ± 0.68 | 85.87 ± 0.73 | 69.55 ± 1.63 | 67.18 ± 1.32 | 65.90 ± 0.62 | 8.00 ± 2.87 |
| Node2vec | 64.20 ± 0.79 | 61.43 ± 0.81 | 69.29 ± 0.70 | 83.15 ± 0.86 | 85.36 ± 0.64 | 70.35 ± 1.44 | 66.94 ± 1.57 | 65.75 ± 0.77 | 7.87 ± 2.36 |
| DGI | **86.05 ± 0.60** | 83.83 ± 0.70 | 90.82 ± 0.65 | 79.06 ± 0.99 | 84.38 ± 0.77 | 79.15 ± 0.94 | 76.33 ± 0.97 | 80.92 ± 0.74 | 5.37 ± 2.95 |
| GRACE | 85.43 ± 0.76 | OOM | OOM | 80.32 ± 0.77 | 87.42 ± 0.46 | 77.88 ± 1.31 | 74.52 ± 0.78 | 79.05 ± 0.75 | 5.00 ± 1.82 |
| GMI | 75.60 ± 0.71 | OOM | OOM | 82.43 ± 0.68 | 85.90 ± 0.60 | 74.41 ± 1.25 | 69.40 ± 1.38 | 72.34 ± 0.70 | 7.16 ± 1.21 |
| Hyper2vec | 71.19 ± 1.01 | 72.36 ± 1.08 | OOT | 76.41 ± 0.92 | 79.57 ± 0.85 | 78.05 ± 1.76 | 71.65 ± 1.54 | 71.48 ± 0.88 | 8.14 ± 1.95 |
| LBSN | 48.68 ± 1.08 | 89.08 ± 0.68 | 63.65 ± 1.60 | 79.43 ± 0.80 | 87.05 ± 0.60 | 74.29 ± 1.64 | 69.63 ± 0.98 | 66.10 ± 0.77 | 7.62 ± 2.34 |
| TriCL | 77.40 ± 0.76 | OOM | OOM | **83.99 ± 0.70** | 87.78 ± 0.44 | 81.96 ± 1.42 | 76.69 ± 0.79 | 80.45 ± 0.67 | 3.66 ± 1.69 |
| **VilLain$_B^{128}$** | 79.39 ± 0.78 | 93.49 ± 0.66 | 92.97 ± 0.60 | 79.76 ± 0.54 | 86.05 ± 0.51 | 82.48 ± 1.29 | 78.92 ± 1.27 | 81.42 ± 0.79 | 3.87 ± 2.08 |
| **VilLain$_B^{256}$** | 79.44 ± 0.68 | 93.90 ± 0.75 | 92.64 ± 0.52 | 79.81 ± 0.75 | 86.35 ± 0.65 | **83.03 ± 1.18** | **79.95 ± 1.35** | 80.69 ± 0.71 | 3.37 ± 1.93 |
| **VilLain** | 81.61 ± 0.52 | **95.12 ± 0.37** | **94.91 ± 0.36** | 83.19 ± 0.56 | **87.79 ± 0.68** | 82.08 ± 1.42 | 78.95 ± 0.79 | **82.79 ± 0.79** | **1.87 ± 0.92** |

**Table 18: Full results on node clustering (in terms of normalized mutual information). VilLain and VilLain$_B$ (see Appendix D) outperform the existing (hyper)graph representation learning methods.**

| Method | DBLP | Trivago | Amazon | Primary | High | Citeseer | Cora | Pubmed | Rank |
|---|---|---|---|---|---|---|---|---|---|
| Deepwalk | 0.70 ± 0.05 | 16.70 ± 0.21 | 7.75 ± 0.03 | 85.15 ± 0.39 | **100.00 ± 0.00** | 14.62 ± 1.73 | 23.87 ± 1.83 | **34.35 ± 0.33** | 6.50 ± 3.60 |
| Node2vec | 0.91 ± 0.03 | 16.96 ± 0.54 | 7.73 ± 0.13 | 83.47 ± 0.70 | **100.00 ± 0.00** | 14.52 ± 0.82 | 23.80 ± 1.21 | 32.83 ± 0.07 | 7.50 ± 3.20 |
| DGI | 16.63 ± 0.01 | 44.50 ± 1.68 | 13.01 ± 0.95 | 84.43 ± 1.68 | 73.88 ± 0.95 | 29.09 ± 0.61 | 32.07 ± 0.79 | 31.27 ± 0.00 | 7.50 ± 2.06 |
| GRACE | 42.96 ± 0.11 | OOM | OOM | 67.59 ± 1.12 | 98.17 ± 0.18 | 33.04 ± 1.33 | 46.04 ± 2.44 | 31.55 ± 0.11 | 6.16 ± 3.02 |
| GMI | 27.80 ± 2.61 | OOM | OOM | 84.08 ± 0.54 | 93.10 ± 0.26 | 25.33 ± 1.72 | 42.60 ± 3.56 | 18.71 ± 0.01 | 8.50 ± 1.25 |
| Hyper2vec | 43.40 ± 0.94 | 66.33 ± 0.27 | OOT | **92.48 ± 0.35** | 99.34 ± 0.30 | 34.28 ± 0.30 | 45.53 ± 1.05 | 33.62 ± 0.13 | **2.57 ± 0.90** |
| LBSN | 1.05 ± 0.00 | 39.41 ± 0.12 | 2.68 ± 0.33 | 85.53 ± 0.55 | 97.80 ± 0.24 | 12.14 ± 0.43 | 28.96 ± 0.30 | 4.62 ± 0.59 | 8.50 ± 1.87 |
| TriCL | 38.00 ± 0.02 | OOM | OOM | 87.83 ± 1.22 | 98.74 ± 0.00 | 34.41 ± 0.02 | 44.75 ± 0.30 | 33.74 ± 0.01 | 3.66 ± 1.37 |
| **VilLain$_B^{128}$** | 35.77 ± 1.92 | 56.51 ± 0.41 | 31.94 ± 0.13 | 89.40 ± 0.04 | 98.72 ± 0.00 | 31.60 ± 0.73 | 44.99 ± 1.84 | 32.40 ± 0.00 | 4.75 ± 1.56 |
| **VilLain$_B^{256}$** | 35.90 ± 0.92 | 58.85 ± 0.40 | 31.23 ± 0.16 | 89.76 ± 1.18 | 98.72 ± 0.00 | 32.34 ± 1.79 | 49.08 ± 1.23 | 33.43 ± 0.02 | 3.62 ± 1.21 |
| **VilLain** | **46.58 ± 0.62** | **69.35 ± 0.32** | **35.24 ± 0.48** | 85.67 ± 1.88 | 98.72 ± 0.00 | **34.53 ± 0.45** | **50.38 ± 2.25** | 32.73 ± 0.00 | 2.62 ± 2.11 |

**Table 19: Full results on node retrieval (in terms of mean average precision). VilLain and VilLain_B (see Appendix D) outperform the existing (hyper)graph representation learning methods.**

| Method | DBLP | Trivago | Amazon | Primary | High | Citeseer | Cora | Pubmed | Rank |
|---|---|---|---|---|---|---|---|---|---|
| Deepwalk | 21.34 ± 0.24 | 7.47 ± 0.11 | 27.71 ± 0.17 | 81.62 ± 0.00 | 98.72 ± 0.00 | 27.61 ± 0.07 | 29.24 ± 0.14 | 48.95 ± 0.19 | 8.37 ± 1.99 |
| Node2vec | 21.61 ± 0.20 | 7.09 ± 0.07 | 27.78 ± 0.17 | 81.07 ± 0.00 | 98.58 ± 0.00 | 27.29 ± 0.05 | 29.42 ± 0.15 | 49.39 ± 0.21 | 8.50 ± 1.65 |
| DGI | 36.06 ± 0.37 | 37.32 ± 0.13 | 31.13 ± 0.38 | 89.73 ± 0.00 | 97.81 ± 0.00 | 43.78 ± 0.10 | 50.64 ± 0.30 | 61.65 ± 0.36 | 4.75 ± 1.29 |
| GRACE | 50.22 ± 0.60 | OOM | OOM | 61.41 ± 0.00 | **99.49** ± 0.00 | 41.09 ± 0.08 | 54.17 ± 0.36 | 60.94 ± 0.36 | 5.33 ± 3.19 |
| GMI | 34.63 ± 0.33 | OOM | OOM | 80.00 ± 0.00 | 97.78 ± 0.00 | 35.89 ± 0.07 | 41.62 ± 0.28 | 55.10 ± 0.31 | 8.33 ± 0.74 |
| Hyper2vec | 35.47 ± 0.41 | 43.11 ± 0.48 | OOT | 85.74 ± 0.00 | 90.70 ± 0.00 | 41.21 ± 0.09 | 46.67 ± 0.27 | 55.62 ± 0.19 | 6.57 ± 2.44 |
| LBSN | 21.01 ± 0.09 | 19.05 ± 0.14 | 29.11 ± 0.37 | 81.30 ± 0.00 | 93.22 ± 0.00 | 30.60 ± 0.09 | 40.09 ± 0.26 | 43.50 ± 0.39 | 8.62 ± 2.05 |
| TriCL | 45.11 ± 0.56 | OOM | OOM | 89.91 ± 0.00 | 97.64 ± 0.00 | 42.37 ± 0.10 | 54.98 ± 0.26 | 61.94 ± 0.25 | 4.66 ± 2.13 |
| **VilLain$_B^{128}$** | 47.27 ± 0.54 | 35.39 ± 0.76 | 50.34 ± 0.53 | 89.65 ± 0.00 | 99.21 ± 0.00 | 43.33 ± 0.10 | 53.57 ± 0.29 | 62.50 ± 0.35 | 3.87 ± 1.05 |
| **VilLain$_B^{256}$** | 53.78 ± 0.58 | 42.36 ± 0.62 | 50.61 ± 0.52 | 90.03 ± 0.00 | 99.22 ± 0.00 | 44.35 ± 0.09 | 56.11 ± 0.30 | 61.51 ± 0.35 | 2.50 ± 1.00 |
| **VilLain** | **60.15** ± 0.55 | **67.23** ± 0.72 | **53.64** ± 0.47 | **91.26** ± 0.00 | 98.99 ± 0.00 | **46.37** ± 0.10 | **57.96** ± 0.27 | **64.43** ± 0.36 | **1.37** ± 0.99 |

**Table 20: Effects of $k$s in node classification (NCS), hyperedge prediction (HP), and node clustering (NCT) node retrieval (NR). VilLain benefits from the long-range propagation during training.**

| | Method | DBLP | Trivago | Amazon | Primary | High | Citeseer | Cora | Pubmed | Rank |
|---|---|---|---|---|---|---|---|---|---|---|
| NCS | $k = 1$ | 74.25 ± 1.05 | 78.14 ± 1.89 | 52.16 ± 2.36 | **94.67** ± 2.47 | **99.51** ± 0.39 | 60.48 ± 3.17 | 74.96 ± 1.04 | 78.21 ± 2.18 | 3.00 ± 1.22 |
| | $k = 2$ | 75.76 ± 1.41 | 78.44 ± 1.84 | 55.09 ± 2.55 | 93.43 ± 3.84 | 99.29 ± 0.39 | 60.17 ± 3.69 | **75.15** ± 1.30 | 78.97 ± 1.38 | 2.62 ± 0.85 |
| | $k = 4$ | 77.16 ± 1.26 | 79.43 ± 1.63 | 57.95 ± 2.47 | 93.66 ± 3.93 | 99.19 ± 0.41 | 61.53 ± 3.17 | 75.03 ± 1.38 | 78.82 ± 1.47 | 2.25 ± 0.43 |
| | $k = 8$ | **78.22** ± 1.20 | **80.24** ± 1.89 | **59.12** ± 2.58 | 92.47 ± 4.00 | 98.78 ± 0.69 | **62.05** ± 3.52 | 74.24 ± 1.49 | **79.22** ± 1.73 | **2.12** ± 1.45 |
| HP | $k = 1$ | 80.72 ± 0.68 | 94.60 ± 0.56 | 93.81 ± 0.48 | **83.51** ± 0.64 | 87.54 ± 0.63 | 81.51 ± 1.28 | 77.23 ± 0.91 | 81.80 ± 0.79 | 3.50 ± 1.00 |
| | $k = 2$ | 81.22 ± 0.61 | 94.82 ± 0.48 | 94.79 ± 0.44 | 83.36 ± 0.72 | 87.69 ± 0.55 | 82.02 ± 1.22 | 78.80 ± 0.90 | 82.29 ± 0.59 | 2.75 ± 0.43 |
| | $k = 4$ | 81.61 ± 0.52 | 95.12 ± 0.37 | 94.91 ± 0.36 | 83.19 ± 0.56 | **87.79** ± 0.68 | 82.08 ± 1.42 | 78.95 ± 0.79 | 82.79 ± 0.79 | 2.00 ± 0.50 |
| | $k = 8$ | **81.97** ± 0.72 | **95.24** ± 0.49 | **95.27** ± 0.26 | 82.99 ± 0.56 | 87.39 ± 0.47 | **82.62** ± 1.13 | **79.13** ± 0.82 | **82.94** ± 0.49 | **1.75** ± 1.29 |
| NCT | $k = 1$ | 43.36 ± 1.66 | 65.28 ± 0.28 | 32.66 ± 0.22 | **90.40** ± 1.86 | 98.72 ± 0.00 | 32.29 ± 2.06 | 44.14 ± 2.38 | **33.96** ± 0.24 | 2.87 ± 1.45 |
| | $k = 2$ | 45.50 ± 1.10 | 67.06 ± 0.52 | 33.13 ± 0.44 | 87.70 ± 1.65 | 98.72 ± 0.00 | 34.16 ± 1.78 | 47.38 ± 2.13 | 33.95 ± 0.27 | 2.50 ± 0.70 |
| | $k = 4$ | 46.58 ± 0.62 | 69.35 ± 0.32 | 35.24 ± 0.48 | 85.67 ± 1.88 | 98.72 ± 0.00 | 34.53 ± 0.45 | 50.38 ± 2.25 | 32.73 ± 0.00 | 2.12 ± 0.59 |
| | $k = 8$ | **48.35** ± 0.06 | **71.44** ± 0.28 | **36.97** ± 0.07 | 84.61 ± 1.19 | 98.72 ± 0.00 | **35.16** ± 0.42 | **50.52** ± 0.94 | 32.52 ± 0.00 | **1.75** ± 1.29 |
| NR | $k = 1$ | 55.55 ± 0.63 | 57.90 ± 0.74 | 48.44 ± 0.52 | 91.07 ± 0.00 | **99.30** ± 0.00 | 43.78 ± 0.11 | 53.84 ± 0.27 | 62.94 ± 0.33 | 3.50 ± 1.00 |
| | $k = 2$ | 58.42 ± 0.61 | 63.12 ± 0.70 | 51.61 ± 0.42 | **91.35** ± 0.00 | 99.10 ± 0.00 | 45.07 ± 0.11 | 55.98 ± 0.27 | 63.60 ± 0.35 | 2.62 ± 0.69 |
| | $k = 4$ | 60.15 ± 0.55 | 67.23 ± 0.72 | 53.64 ± 0.47 | 91.26 ± 0.00 | 98.99 ± 0.00 | 46.37 ± 0.10 | 57.96 ± 0.27 | **64.43** ± 0.36 | **1.87** ± 0.59 |
| | $k = 8$ | **62.75** ± 0.48 | **69.08** ± 0.58 | **56.01** ± 0.64 | 90.69 ± 0.00 | 98.70 ± 0.00 | **47.68** ± 0.10 | 57.72 ± 0.25 | 63.64 ± 0.37 | 2.00 ± 1.22 |

**Table 21: Effects of $k'$s in node classification (NCS), hyperedge prediction (HP), and node clustering (NCT) node retrieval (NR). VilLain benefits from the long-range propagation at inference (i.e., embedding generation).**

| | Method | DBLP | Trivago | Amazon | Primary | High | Citeseer | Cora | Pubmed | Rank |
|---|---|---|---|---|---|---|---|---|---|---|
| **NCS** | $k'=1$ | $64.71_{\pm1.98}$ | $60.60_{\pm1.54}$ | $48.46_{\pm3.45}$ | $\mathbf{96.74}_{\pm0.79}$ | $\mathbf{99.58}_{\pm0.20}$ | $60.62_{\pm2.94}$ | $74.68_{\pm1.57}$ | $77.94_{\pm1.84}$ | $5.62_{\pm2.86}$ |
| | $k'=2$ | $65.29_{\pm1.91}$ | $61.59_{\pm1.62}$ | $49.42_{\pm2.76}$ | $96.36_{\pm2.19}$ | $99.57_{\pm0.21}$ | $60.44_{\pm3.03}$ | $74.70_{\pm1.58}$ | $78.18_{\pm1.75}$ | $5.50_{\pm2.29}$ |
| | $k'=4$ | $66.64_{\pm2.19}$ | $63.25_{\pm1.63}$ | $50.52_{\pm2.45}$ | $96.33_{\pm1.99}$ | $99.39_{\pm0.21}$ | $60.54_{\pm3.28}$ | $74.77_{\pm1.75}$ | $78.29_{\pm2.14}$ | $5.00_{\pm1.58}$ |
| | $k'=8$ | $67.88_{\pm1.79}$ | $65.08_{\pm1.55}$ | $53.22_{\pm3.74}$ | $93.91_{\pm2.57}$ | $99.26_{\pm0.38}$ | $61.29_{\pm3.30}$ | $\mathbf{75.06}_{\pm1.44}$ | $78.75_{\pm1.91}$ | $4.50_{\pm1.41}$ |
| | $k'=16$ | $70.83_{\pm1.70}$ | $68.28_{\pm1.38}$ | $54.65_{\pm3.63}$ | $94.49_{\pm3.05}$ | $98.86_{\pm0.79}$ | $61.62_{\pm3.28}$ | $74.86_{\pm1.34}$ | $79.12_{\pm1.44}$ | $3.75_{\pm0.82}$ |
| | $k'=32$ | $73.20_{\pm1.60}$ | $72.31_{\pm1.65}$ | $55.80_{\pm2.41}$ | $92.57_{\pm3.78}$ | $98.59_{\pm1.42}$ | $61.96_{\pm3.50}$ | $74.68_{\pm1.30}$ | $79.22_{\pm1.69}$ | $4.00_{\pm1.65}$ |
| | $k'=64$ | $76.47_{\pm1.30}$ | $76.77_{\pm1.71}$ | $56.42_{\pm2.59}$ | $94.21_{\pm3.34}$ | $98.50_{\pm2.31}$ | $62.42_{\pm2.93}$ | $74.25_{\pm1.99}$ | $78.98_{\pm1.68}$ | $4.00_{\pm2.29}$ |
| | $k'=128$ | $\mathbf{77.62}_{\pm1.26}$ | $\mathbf{80.63}_{\pm1.36}$ | $\mathbf{57.46}_{\pm2.04}$ | $88.68_{\pm4.90}$ | $98.09_{\pm1.90}$ | $\mathbf{63.67}_{\pm3.26}$ | $74.41_{\pm1.76}$ | $\mathbf{79.37}_{\pm1.92}$ | $3.50_{\pm3.24}$ |
| **HP** | $k'=1$ | $77.77_{\pm0.57}$ | $91.46_{\pm0.57}$ | $93.11_{\pm0.56}$ | $82.05_{\pm0.92}$ | $87.45_{\pm0.65}$ | $82.14_{\pm0.97}$ | $79.63_{\pm0.84}$ | $82.09_{\pm0.70}$ | $6.75_{\pm1.92}$ |
| | $k'=2$ | $77.86_{\pm0.57}$ | $91.53_{\pm0.61}$ | $93.39_{\pm0.75}$ | $82.30_{\pm0.56}$ | $\mathbf{87.74}_{\pm0.61}$ | $82.39_{\pm1.59}$ | $79.29_{\pm0.87}$ | $82.27_{\pm0.56}$ | $5.25_{\pm1.98}$ |
| | $k'=4$ | $78.58_{\pm0.86}$ | $92.51_{\pm0.82}$ | $93.77_{\pm0.77}$ | $82.89_{\pm0.84}$ | $87.53_{\pm0.55}$ | $81.79_{\pm1.24}$ | $78.28_{\pm0.79}$ | $82.12_{\pm0.82}$ | $5.87_{\pm1.83}$ |
| | $k'=8$ | $79.06_{\pm0.83}$ | $92.31_{\pm0.61}$ | $94.07_{\pm0.51}$ | $82.86_{\pm0.79}$ | $87.69_{\pm0.66}$ | $82.23_{\pm1.19}$ | $78.35_{\pm0.93}$ | $82.46_{\pm0.65}$ | $5.00_{\pm1.11}$ |
| | $k'=16$ | $79.87_{\pm0.63}$ | $92.97_{\pm0.44}$ | $94.59_{\pm0.39}$ | $83.06_{\pm0.71}$ | $87.55_{\pm0.62}$ | $82.16_{\pm0.97}$ | $78.87_{\pm0.89}$ | $82.49_{\pm0.75}$ | $4.12_{\pm1.46}$ |
| | $k'=32$ | $80.39_{\pm0.53}$ | $93.80_{\pm0.60}$ | $95.15_{\pm0.38}$ | $82.89_{\pm0.84}$ | $87.28_{\pm0.45}$ | $82.74_{\pm1.15}$ | $79.29_{\pm0.84}$ | $82.92_{\pm0.65}$ | $3.25_{\pm1.56}$ |
| | $k'=64$ | $81.06_{\pm0.47}$ | $94.58_{\pm0.61}$ | $95.15_{\pm0.42}$ | $82.78_{\pm0.85}$ | $87.70_{\pm0.44}$ | $83.03_{\pm0.92}$ | $79.51_{\pm0.78}$ | $82.75_{\pm0.55}$ | $\mathbf{2.62}_{\pm0.99}$ |
| | $k'=128$ | $\mathbf{81.89}_{\pm0.87}$ | $\mathbf{95.15}_{\pm0.49}$ | $\mathbf{95.21}_{\pm0.36}$ | $81.91_{\pm0.77}$ | $87.04_{\pm0.70}$ | $\mathbf{83.42}_{\pm1.19}$ | $\mathbf{80.23}_{\pm0.85}$ | $\mathbf{83.00}_{\pm0.55}$ | $2.75_{\pm3.03}$ |
| **NCT** | $k'=1$ | $29.62_{\pm1.44}$ | $48.38_{\pm0.50}$ | $31.74_{\pm0.05}$ | $93.08_{\pm0.05}$ | $\mathbf{98.72}_{\pm0.00}$ | $33.04_{\pm1.07}$ | $48.93_{\pm2.07}$ | $32.36_{\pm0.00}$ | $5.25_{\pm2.72}$ |
| | $k'=2$ | $30.30_{\pm1.57}$ | $49.05_{\pm0.33}$ | $31.74_{\pm0.40}$ | $93.09_{\pm0.01}$ | $98.72_{\pm0.00}$ | $33.79_{\pm0.53}$ | $49.78_{\pm1.39}$ | $32.30_{\pm0.00}$ | $4.87_{\pm2.75}$ |
| | $k'=4$ | $30.26_{\pm1.88}$ | $50.24_{\pm0.32}$ | $31.81_{\pm0.45}$ | $91.95_{\pm1.25}$ | $98.72_{\pm0.00}$ | $33.95_{\pm0.78}$ | $48.40_{\pm1.85}$ | $32.31_{\pm0.48}$ | $4.87_{\pm1.76}$ |
| | $k'=8$ | $30.39_{\pm1.45}$ | $51.75_{\pm0.75}$ | $33.62_{\pm0.03}$ | $86.48_{\pm1.53}$ | $98.72_{\pm0.00}$ | $34.78_{\pm0.54}$ | $\mathbf{50.74}_{\pm1.88}$ | $32.77_{\pm0.01}$ | $3.50_{\pm1.73}$ |
| | $k'=16$ | $31.90_{\pm1.66}$ | $54.59_{\pm0.27}$ | $34.29_{\pm0.05}$ | $84.45_{\pm1.26}$ | $98.72_{\pm0.00}$ | $34.97_{\pm0.68}$ | $48.65_{\pm1.49}$ | $32.61_{\pm0.00}$ | $3.62_{\pm1.11}$ |
| | $k'=32$ | $35.63_{\pm1.87}$ | $58.75_{\pm0.66}$ | $34.05_{\pm0.46}$ | $83.87_{\pm0.32}$ | $98.42_{\pm0.19}$ | $35.54_{\pm1.93}$ | $47.49_{\pm0.47}$ | $32.98_{\pm0.00}$ | $4.25_{\pm2.10}$ |
| | $k'=64$ | $44.85_{\pm1.29}$ | $64.91_{\pm0.36}$ | $35.40_{\pm0.58}$ | $84.87_{\pm0.32}$ | $97.65_{\pm0.00}$ | $37.41_{\pm0.76}$ | $45.56_{\pm1.05}$ | $32.65_{\pm0.00}$ | $3.87_{\pm2.52}$ |
| | $k'=128$ | $\mathbf{47.62}_{\pm0.05}$ | $\mathbf{70.88}_{\pm0.30}$ | $\mathbf{35.66}_{\pm0.61}$ | $83.92_{\pm0.38}$ | $98.09_{\pm0.00}$ | $36.68_{\pm0.22}$ | $44.35_{\pm0.87}$ | $32.12_{\pm0.00}$ | $4.37_{\pm3.15}$ |
| **NR** | $k'=1$ | $44.77_{\pm0.54}$ | $31.08_{\pm0.79}$ | $51.00_{\pm0.42}$ | $91.94_{\pm0.00}$ | $\mathbf{99.41}_{\pm0.00}$ | $45.47_{\pm0.11}$ | $57.46_{\pm0.26}$ | $63.04_{\pm0.33}$ | $5.50_{\pm2.39}$ |
| | $k'=2$ | $45.34_{\pm0.55}$ | $30.95_{\pm0.82}$ | $50.25_{\pm0.41}$ | $91.95_{\pm0.00}$ | $99.35_{\pm0.00}$ | $45.62_{\pm0.11}$ | $57.51_{\pm0.26}$ | $63.15_{\pm0.34}$ | $5.25_{\pm2.33}$ |
| | $k'=4$ | $46.06_{\pm0.60}$ | $33.43_{\pm0.83}$ | $48.84_{\pm0.45}$ | $91.72_{\pm0.00}$ | $99.22_{\pm0.00}$ | $45.05_{\pm0.10}$ | $56.59_{\pm0.27}$ | $63.21_{\pm0.34}$ | $5.75_{\pm2.04}$ |
| | $k'=8$ | $48.00_{\pm0.57}$ | $36.56_{\pm0.77}$ | $51.32_{\pm0.40}$ | $91.41_{\pm0.00}$ | $99.06_{\pm0.00}$ | $46.25_{\pm0.10}$ | $57.79_{\pm0.26}$ | $63.46_{\pm0.35}$ | $4.25_{\pm0.96}$ |
| | $k'=16$ | $50.65_{\pm0.58}$ | $40.19_{\pm0.65}$ | $51.19_{\pm0.39}$ | $91.01_{\pm0.00}$ | $98.79_{\pm0.00}$ | $46.77_{\pm0.10}$ | $58.04_{\pm0.26}$ | $\mathbf{63.50}_{\pm0.36}$ | $4.00_{\pm1.22}$ |
| | $k'=32$ | $54.32_{\pm0.59}$ | $46.72_{\pm0.76}$ | $\mathbf{54.67}_{\pm0.37}$ | $90.89_{\pm0.00}$ | $98.49_{\pm0.00}$ | $47.09_{\pm0.10}$ | $58.23_{\pm0.25}$ | $63.31_{\pm0.37}$ | $3.37_{\pm1.65}$ |
| | $k'=64$ | $58.34_{\pm0.58}$ | $60.01_{\pm0.77}$ | $53.54_{\pm0.48}$ | $90.88_{\pm0.00}$ | $98.33_{\pm0.00}$ | $\mathbf{47.10}_{\pm0.10}$ | $\mathbf{58.29}_{\pm0.26}$ | $62.97_{\pm0.36}$ | $3.62_{\pm2.64}$ |
| | $k'=128$ | $\mathbf{61.23}_{\pm0.53}$ | $\mathbf{70.75}_{\pm0.79}$ | $52.60_{\pm0.42}$ | $90.88_{\pm0.00}$ | $98.29_{\pm0.00}$ | $46.80_{\pm0.10}$ | $58.28_{\pm0.26}$ | $62.76_{\pm0.36}$ | $4.12_{\pm2.84}$ |

