# OpenReview forum: "VilLain: Self-Supervised Learning on Hypergraphs without Features via Virtual Label Propagation"
_ACM.org/TheWebConf/2024/Conference — TheWebConf24_

### Official Review · Reviewer_EHvH · 2023-10-29

**Novelty:** 6
**Technical Quality:** 6

**Review:**

### Summary

The paper introduces VilLain, a self-supervised learning method, to address the challenge of learning node representations in hypergraphs without external input features or output labels.

VilLain utlises virtual labels (v-labels) and propagates sparse probability distributions over these labels as feature vectors for nodes.

VilLain stands out for its requirement-free approach, providing versatile and accurate embeddings applicable to a wide range of tasks, including node classification, node retrieval, and hyperedge prediction.


$~$

### Quality

The motivating observations (Section 4) and the methodology (Section 5) demonstrate soundness in tackling the intricate challenge of learning node representations devoid of external output labels or input features.

The devised loss functions for v-labels are intricately tailored to match the higher-order label homogeneity patterns observed (i.e., Observations 1 and 2).

The experimental design, including tasks like node classification, hyperedge prediction, node clustering, and node retrieval, highlights the versatility of the proposed approach.


$~$

### Clarity

The clarity of loss terms to capture global information from lines 387 to 445 could be enhanced by providing intuitive explanations for each step in these terms.

For instance, motivating the choice of cosine similarity in Equation 6, and discussing why $l_1$ normalisation is used in Equation 5 but softmax normalisation in Equations 6 and 1, would enhance clarity.

Additionally, there needs to be a clearer explanation for the use of the double logarithm in the random noise term $g_j$ (or is it $g_i$?) on line 308.


$~$

### Originality

The self-supervised loss functions delve into structural intricacies and extend further, leveraging higher-order label homogeneity.

The approach to input feature learning represents a departure from previous techniques, demonstrating originality.

Overall, the paper situates itself within the realm of related research.

$~$

### Significance

The significance lies in the applicability of VilLain's general-purpose node embeddings to diverse downstream tasks.

It is interesting to see that the proposed method is also effective on datasets with external node features (as shown in Table 10).

The discussion of VilLain's limitation, especially in the context of hypergraphs displaying heterophilic traits with potentially diminished label homogeneity, underscores the necessity for future advancements.


$~$

### Pros

\+ Sound, Tailored Approach

\+ Versatile Experimental Design

\+ Original Approach


$~$

### Cons

\- Clarity Enhancement Needed

\- Explanation Depth Lacking

$~$

**Questions:**

1. What was the rationale behind choosing cosine similarity in Equation 6 for distinctiveness in the context of capturing global information?
2. Why was $l_1$ normalisation chosen in Equation 5 while softmax normalisation was employed in Equations 6 and 1?
3. What advantages did these specific normalisation techniques offer in their respective contexts?
4. What could be an intuitive understanding of the usage of the double logarithm in the random noise term?
5. What led to the final decision to employ these specific choices in all the previous questions?

**Reviewer Confidence:**

3: The reviewer is confident but not certain that the evaluation is correct

**Scope:**

4: The work is relevant to the Web and to the track, and is of broad interest to the community

---

### Official Review · Reviewer_kgVD · 2023-11-16

**Novelty:** 4
**Technical Quality:** 3

**Review:**

This paper presents a novel self-supervised hypergraph representation learning method, called VilLain, which is based on the propagation of virtual labels (v-labels). VilLain is Requirement-free (without relying on node labels and features), Versatile (generalizable to diverse downstream tasks), and Accurate. VilLain generates v-labels for node features, and optimizes them while propagating them along the hypergraph structure.

Pros:
* The paper clearly justifies the observations about real-world hypergraphs (label homogeneity and higher-order label homogeneity), which motivate the design of VilLain.
* The paper describes the methodology and experiment settings in detail. It conducts extensive experiments to evaluate the performance of VilLain and to perform ablation study and parameter study.
* VilLain achieves the best performance on four downstream tasks.

Cons:
* Fig.3 fails to present the overall framework of VilLain well. Some important elements such as loss functions are not included in this figure, which makes it hard to understand the overall framework of ViLain. A more detailed figure or a simple pseudo-code Algorithm can help describe VilLain better.
* Many baselines are outdated, especially in tasks of Hyperedge Prediction, Node Clustering, and Node Retrieval. In these three tasks, most of the baselines are at least three years old except for TriCL. Some more recent baselines can be considered while some earlier baselines can be removed. For example, LHQD [1], CIAH [2], and HEM [3] can be considered for the task of node clustering.
* Some typos exist, such as in Section Introduction at line 123, "three downstream tasks" should be "four downstream tasks". In Section Reference, pages of references should be included.


[1] Liu, Meng, et al. Strongly local hypergraph diffusions for clustering and semi-supervised learning. 2021 Proceedings of the Web Conference (2021), 2092-2103.
[2] Yang, T., Yang, C., Zhang, L., Shi, C., Hu, M., Liu, H., ... & Wang, D. 2022. Co-clustering interactions via attentive hypergraph neural network. In Proceedings of the 45th International ACM SIGIR Conference on Research and Development in Information Retrieval, (2022) 859-869.
[3] Feng, Z., Qiao, M., & Cheng, H. Modularity-based Hypergraph Clustering: Random Hypergraph Model, Hyperedge-cluster Relation, and Computation. 2023. Proceedings of the ACM on Management of Data, 1(3), 1-25.

**Questions:**

* Can the authors add some recent baselines?
* It seems VilLain does not rely on any neural network module. While so many neural network models are reported to have great representation ability, how does VilLain outperform them without neural networks?

**Reviewer Confidence:**

3: The reviewer is confident but not certain that the evaluation is correct

**Scope:**

4: The work is relevant to the Web and to the track, and is of broad interest to the community

---

### Official Review · Reviewer_PDG4 · 2023-11-26

**Novelty:** 5
**Technical Quality:** 5

**Review:**

This work proposes a new self-supervised learning hypergraph self-supervised learning method called VilLan based on the propagation of the virtual labels. Specifically, it first learns a sparse probability distribution over the virtual labels as the feature vector in hypergraph learning. Then it propagates the vectors to learn the final embeddings. This method is requirement-free, versatile, and effective. Extensive experiments on various benchmark datasets for different downstream tasks show the effectiveness. To conclude, I would like to suggest weakly accepting this work for the following reasons:


Strength:

1. The motivation for requirement-free (feature-free) hypergraph self-supervised learning is clear and impressive. This is an impressive topic in graph contrastive learning. In some extreme cases, obtaining any attribute features for nodes is very hard and even impossible, such as some biology and chemistry graphs.

2. This work is well-written with clear motivations, nice figures, and comprehensive experiments.

3. Experiments over benchmark datasets and an external website dataset for different downstream tasks (i.e., node classification, hyperedge prediction, hypergraph extraction, and recommendation) are comprehensive.

Weakness:

1. I am sort of curious about why this work proposes self-supervised learning with attribute feature-free in hypergraph learning. I thought the problem or we can say, challenges, also exists in graph learning. Why do not first apply the proposed method to graph representation learning?

2. Concerning the Observations 1 and 2, I hold different opinions on these two statements. Sometimes hypergraph also fails to keep the homophily within the dataset. For instance, some nodes within the same hyperedge will belong to different classes according to different downstream tasks. Unlike graphs that can keep the homophily within the graph datasets, sometimes hypergraphs are not able to handle the homophily in a professional way. This would be my major concerns about the model design.

3. The format of some equations should be revised. For instance, a full stop should be added to Equation 7.

**Questions:**

Please refer to the weakness for questions.

**Reviewer Confidence:**

4: The reviewer is certain that the evaluation is correct and very familiar with the relevant literature

**Scope:**

3: The work is somewhat relevant to the Web and to the track, and is of narrow interest to a sub-community

---

### Official Review · Reviewer_vbmt · 2023-11-27

**Novelty:** 5
**Technical Quality:** 6

**Review:**

Summary

This work proposes a method for learning node embeddings for hypergraphs without features. It creates feature vector for each node by soft labels and then propagation. Then loss functions are designed based on the propagated soft labels, for both local and global information.

Strength
- The proposed method does not need node features.
- On 4 categories of tasks, the proposed method outperforms baselines significantly and the ablaiton models. Rich experimental results for different settings and targets.
- The proposed method works for unobserved nodes.

Weakness
- The complexity of the proposed method is still high compared to the SOTA in HGNN [1], which does not need the expensive matrix product with H but still considers high-order interactions.
- The complexity analysis is not complete. It did not discuss the complexity of computing $\mathcal{L}_{global}$.

[1] https://arxiv.org/abs/2306.02560

**Questions:**

- Why is L1 norm used in Eq.(5) instead of L2 or others?
- Why is there no weight for each term in the loss?
- L459, I do not agree that node classification is less structure-dependent.
- Why would not using large k' result in over smoothing?
- L389, what does v labels are properly distributed mean?
- Based on results in Fig. 6, it seems one should always maximize d and d/D, is this true?

**Reviewer Confidence:**

3: The reviewer is confident but not certain that the evaluation is correct

**Scope:**

4: The work is relevant to the Web and to the track, and is of broad interest to the community

---

### Official Review · Reviewer_8Yt4 · 2023-11-28

**Novelty:** 5
**Technical Quality:** 4

**Review:**

This paper focuses on hypergraph representation learning, considering how to conduct representation learning in the absence of labels or features. Specifically, the authors propose using the propagation of virtual labels to construct node features and design a self-supervised learning loss function based on the label homogeneity of hypergraphs, enhancing node representations and achieving good performance in multiple downstream tasks.

Strengths:
1. The proposed method is simple and effective, demonstrating good performance in downstream tasks.
2. It has a relatively positive impact on advancing hypergraph representation learning.

Weaknesses:
1. The writing could be clarified in places, with more explanation around the motivation and how the specific solutions follow from the problem definition. A clearer logical flow would make the overall contributions more evident.
2. Some key claims are not well justified by experiments.

**Questions:**

1. The expressions throughout the paper lack professional refinement, which hinders reader comprehension.
2. There are some symbol errors that need to be reviewed. For example, Section Multi-V-label Propagation, the formula for Z is incorrect.
3. The homogeneity analysis of hypergraphs in Section 4 lacks deeper insights and seems to present commonly known conclusions.
4. The motivation behind using multiple virtual labels when nodes do not have labels or features is unclear in the paper.
5. In the node classification task, do the semi-supervised/supervised methods directly follow the training methods of the original model?
6. What is the calculation method for NMI, and what aspect of clustering quality does it measure?
7. The analysis of L_global is insufficient and should verify the respective effects of cls and dst.
8. Although the authors stated not to add a balancing hyperparameter between L_local and L_global, results regarding this parameter can be provided in the experiments to demonstrate more meaningful conclusions.
9. I believe that "Requirement-free" and "Versatile" cannot be regarded as contributions or advantages of the proposed method in this paper, as other unsupervised methods also possess these characteristics.
10. Is there any necessary connection between the designed method and the hypergraph scene? Is it also applicable in ordinary graph learning, why do it on the hypergraph?
11. Can you explain the connection and difference between the loss function proposed in this article and the alignment/uniformity loss functions in contrastive learning?

**Reviewer Confidence:**

4: The reviewer is certain that the evaluation is correct and very familiar with the relevant literature

**Scope:**

4: The work is relevant to the Web and to the track, and is of broad interest to the community

---

### Decision · Program_Chairs · 2024-01-22

**Decision:**

Accept

**Comment:**

**Meta-review**: The paper introduces VilLain, a self-supervised learning method, to address the challenge of learning node representations in hypergraphs without external input features or output labels. While many unsupervised and featureless methods (e.g. DeepWalk) exist for regular graphs, most prior work in hyper-graphs has focused on methods that include features. While reviewers generally liked the featureless setting, I feel like the novelty here could be overstated (for example, it is generally easy to either manufacture some features or use trivial features like the identity matrix if required). However, reviews are generally positive, and the authors do a good job of responding to questions and weaknesses. Looks like an accept.

 **Strengths**:
 + Effectiveness of method (8Yt4, vbmt, PDG4, kgVD)
 + Novelty of featureless method (vbmt, PDG4)

 **Weaknesses**: *most addressed during discussion*
 - Older baselines (kgVD)